# Functional and analytical recapitulation of osteoclast biology on demineralized bone paper

Yongkuk Park ●[1], Tadatoshi Sato[2] & Jungwoo Lee ●[1,3,4] ✉

Osteoclasts are the primary target for osteoporosis drug development. Recent animal studies revealed the crucial roles of osteoblasts in regulating osteoclastogenesis and the longer lifespans of osteoclasts than previously thought with fission and recycling. However, existing culture platforms are limited to replicating these newly identified cellular processes. We report a demineralized bone paper (DBP)-based osteoblast culture and osteoclast assay platform that replicates osteoclast fusion, fission, resorption, and apoptosis with high fidelity and analytical power. An osteoid-inspired DBP supports rapid and structural mineral deposition by osteoblasts. Coculture osteoblasts and bone marrow monocytes under biochemical stimulation recapitulate osteoclast differentiation and function. The DBP-based bone model allows longitudinal quantitative fluorescent monitoring of osteoclast responses to bisphosphonate drug, substantiating significantly reducing their number and lifespan. Finally, we demonstrate the feasibility of humanizing the bone model. The DBP-based osteo assay platforms are expected to advance bone remodeling-targeting drug development with improved prediction of clinical outcomes.

In the United States, over 10 million people suffer from osteoporosis while another 40 million are diagnosed with osteopenia, a precursor condition to osteoporosis characterized by significantly reduced bone mineral density. About 2 million osteoporotic fractures occur each year. Nearly 20% of these patients die within a year because of additional fractures and complications[1–3]. Standard care focuses on the pharmacological prevention of bone loss using antiresorptive and anabolic drugs. However, these drugs are limited for long-term use due to the increased risk of adverse effects[4–6]. Discontinuing treatment often leads to rapid and substantial bone loss that is difficult to regain[7,8]. Therefore, an in-depth understanding of bone metabolic regulation and developing new osteoporosis drugs are urgently needed for the long-term safe and effective management of this debilitating condition. Achieving these goals requires the development of functional and analytical

in vitro assays that can accurately replicate in vivo-relevant bone remodeling processes.

Repeated bone remodeling by the balanced action between bone-forming osteoblasts and bone-breaking osteoclasts is an essential physiological process for maintaining mechanical structure and mineral homeostasis. Previous studies identified that the receptor activator of nuclear factor kappa-B ligand (RANKL) and its decoy receptor, osteoprotegerin (OPG), are key molecular regulators of bone remodeling (Fig. 1A)[9–11]. This discovery led to the recapitulation of osteoclastogenesis in vitro by adding RANKL and macrophage-colony stimulating factor (M-CSF) in bone marrow monocyte (BMM) culture[12]. BMMs on tissue culture plastic (TCP) continually fuse to form multinucleated giant osteoclasts until apoptotic death[13]. However, recent intravital imaging studies of mouse calvaria and tibia have revealed that osteoclastogenesis is more complex than previously observed in vitro, with at least two distinct

[1]Department of Chemical Engineering, Institute for Applied Life Sciences, University of Massachusetts, Amherst, MA 01003, USA. [2]Department of Medicine, UMass Chan Medical School, Worcester, MA 01605, USA. [3]Department of Biomedical Engineering, University of Massachusetts, Amherst, MA 01003, USA. [4]Molecular & Cellular Biology Graduate Program, University of Massachusetts, Amherst, MA 01003, USA. ✉e-mail: jungwoo@umass.edu

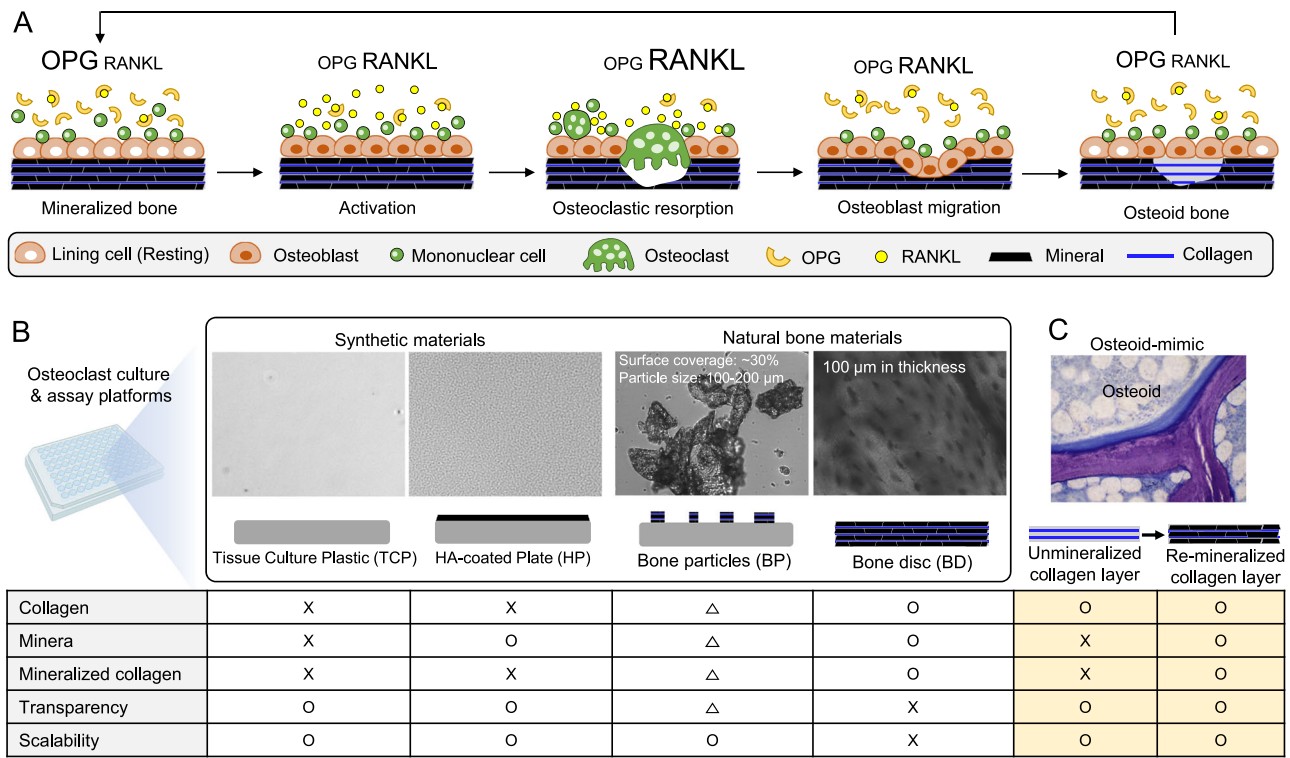

**Fig. 1 | Current understanding of bone remodeling cycle and osteoclast assay platforms. A** Receptor activator of nuclear factor kappa-B ligand (RANKL) and Osteoprotegerin (OPG) are key regulatory molecules for bone remodeling. RANKL binding to a RANK receptor on bone marrow monocytes triggers their differentiation into multinucleated osteoclasts, whereas OPG binding to RANKL blocks RANKL-RANK signaling in bone marrow monocytes (BMMs) and inhibits their osteoclast differentiation. A bone remodeling cycle starts with the activation of osteoblasts that increases RANKL but decreases OPG secretion to induce osteo-clastogenesis of BMMs. As the osteoclasts break down the existing bone matrix, osteoblasts surround the dissolved bone area and initiate new bone formation, reducing RANKL but increasing OPG secretion. At the end of bone formation, osteoblasts become resting-state while secreting high OPG to prevent unnecessary bone remodeling. **B** Current osteoclast assay platforms can be divided into tissue culture plastic (TCP)-based synthetic and extracellular matrix (ECM)-based substrates. Each substrate represents advantages and disadvantages in mimicking bone ECM and osteoclast assays. The 96-well plate image has been obtained from BioRender.com. **C** We propose an osteoid-inspired demineralized bone paper (DBP) as an enabling biomaterial platform for in vitro osteoclast differentiation and functional assay that represents competitive advantages over the existing platforms. (o applicable, Δ partially applicable, x not applicable).

aspects[14]. First, osteoblasts play a critical role in regulating osteoclast differentiation and function via secreted molecules and direct contact. For example, osteoblasts secrete RANKL and OPG but in a different profile depending on bone metabolic state[15]. Osteoclasts that contact with osteoblasts exhibited significantly reduced bone resorption activity compared to osteoclasts that are not in contact with osteoblasts[16]. Second, osteoclasts can live significantly longer than previously expected. A subset of osteoclasts does not undergo apoptotic death at the end of bone resorption but separates into individual monocytes via fission, which can reassemble into osteoclasts in subsequent bone remodeling cycles[17]. Recapitulating these newly identified processes of osteoclasts in vitro presents a new challenge for accurately identifying and developing effective osteoclast-targeting drugs.

The appropriate selection of biomaterial substrates is critical for recapitulating in vivo-relevant osteoclast differentiation and function since osteoclasts exclusively develop and function on the mineralized bone surface. While TCP supports reproducible and analytical osteo-clastogenesis, it cannot reproduce their bone resorption function. Alternatively, hydroxyapatite-coated TCP (HP) allows osteoclastic mineral resorption[18–21]. However, osteoclasts on TCP and HP exhibit abnormally giant morphology and undergo apoptosis in 1-2 weeks, contradicting recent in vivo discoveries. Bone discs (BD) have been considered the gold-standard for osteoclast assays because they support in vivo-relevant osteoclast differentiation of BMMs and bone resorption[22–25]. However, the laborious preparation and opaqueness of BD have restricted their broad usage and application. Bone particle-

coated TCP (BP) has been introduced as an intermediate solution, which allows for microscopic monitoring of osteoclastic bone resorption by measuring decreasing bone particle sizes[26–28]. However, non-standardized bone particles and the co-arising osteoclasts on TCP and bone particles remain critical drawbacks to establishing controlled and reproducible osteoclast assays. Currently, there is no widely accepted and capable biomaterial substrate for in vitro osteoclastogenesis and functional assays in a scalable and analytic manner (Fig. 1B).

Here, we introduce an osteoclast differentiation and functional assay platform based on demineralized bone paper (DBP), a thin slice of demineralized compact bone matrix. Osteoblastic bone formation begins with the synthesis of a collagen-based structural framework, known as osteoid, followed by mineral deposition (Fig. 1C)[29]. We hypothesized that a thin section of demineralized compact bone preserving intact collagen structure functions as an osteoid mimic to recapitulate in vivo-relevant bone tissue complexity and associated multicellular processes. DBP induces rapid and structural mineral deposition by murine osteoblasts, while remineralized-DBP retains sufficient transparency for microscopic imaging. DBP is also scalable for production and mechanically durable for easy experimental handling. By exploiting these enabling features, we first demonstrated longitudinal fluorescent monitoring of osteoclastogenesis of murine BMMs and subsequent bone resorption. We then recapitulated a bone remodeling cycle, including osteoclast fusion, bone resorption, and fission, by stimulating osteoblasts and BMMs coculture on DBP with vitamin D3 and prostaglandin E2. We also simulated osteoclast-targeting bisphosphonate drug action

on the coculture model using a fluorescent dye-conjugated bisphosphonate. Finally, we demonstrated the feasibility of humanizing the DBP-based bone model with human osteoblasts differentiated from bone marrow-derived stromal cells and human CD14+ monocytes derived from peripheral blood that differentiate into osteoclasts. The established osteoclast culture and assay platform will greatly facilitate bone remodeling targeting osteoporosis drug development.

## Results

### Demineralized bone paper mimics osteoid, unmineralized collagen-based bone matrix

We generated DBP using bovine femurs[29]. Femoral bones were cut into 4–6 cm length blocks and cleaned by removing periosteal connective tissue and inner marrow. Minerals were dissolved by hydrochloric acid (1.2 N) under cyclic hydrostatic pressure (4 bar, 0.1 Hz). Demineralization status was checked by X-ray scanning. If residual minerals are detected, we repeat the demineralization process. Complete demineralization took 5–7 days. We sliced a fully demineralized bone block into 5–200 μm thickness using a cryostat (Fig. 2A). Depending on the sectioning direction, DBP exhibited distinct collagen patterns: transverse section showed concentric lamellae of an osteon and vertical section displayed parallel lamellae (Fig. 2B). The biochemical integrity of the collagen matrix was confirmed by a fluorescent dye-conjugated collagen-hybridizing peptide (CHP) that selectively binds to denatured collagen fibrils[30] and multiphoton microscopy that visualizes intact triple-helical collagen structure by second harmonic generation (SHG)[31]. Heat-treated DBP with denatured collagen displayed strong CHP fluorescence and weak SHG signal, whereas control DBP showed no CHP fluorescence but a strong SHG signal. These results indicate that the collagen fibers in DBP remain biochemically intact (Fig. 2C). Finally, we removed residual cellular materials from DBP by treating 1% sodium dodecyl sulfate (SDS), which was confirmed by nuclear 4′,6-diamidino2-phenylindole (DAPI) staining (Fig. 2D).

Next, we demonstrated the manipulation of DBP thickness (20–200 μm) and characterized its optical and mechanical properties. As expected, decreasing DBP thickness increased optical transparency (Fig. 2E) and decreased mechanical stiffness (Fig. 2F). A 20 μm thick DBP showed 90% light transmittance while retaining mechanical durability for easy experimental handling. In this study, we used a 20 μm thick vertically sliced DBP, which represents better the collagen structure of osteoid than transversely sliced DBP. A typical DBP was larger than 2 × 2 cm. To fit the DBP into a 96-well plate, we punched it into a circular shape using a biopsy punch (D = 6 mm) (Fig. 2G). DBP can be produced in large quantities, more than 50,000 from a single bovine femur, sufficient to prepare 520 of 96-well plate assays. These results indicate that DBP retains the intrinsic collagen matrix structure and complexity that closely resembles an unmineralized structural collagen matrix in osteoid.

### DBP induces rapid and structural mineral deposition by osteoblasts

We conducted a comparative test to determine whether DBP provides better support for in vivo-relevant mineralized bone formation than existing osteogenic assay platforms, including TCP, hydroxyapatite-coated plate (HP), bone particle-coated plate (BP), and bovine bone disc (BD). To monitor osteogenic and hematopoietic cells over the long term, we used DsRed and eGFP reporter mice in this study. We retrieved osteogenic cells from femoral bone chips of DsRed mice and expanded them before use. We seeded osteoblasts on 5 different substrates and cultured for 1 week in osteogenic differentiation medium. We then characterized cell morphology and proliferation (Fig. 3A). Actin (phalloidin) staining provided insight into the cytoskeletal organization and morphology of the osteoblasts on each surface. Osteoblasts cultured on DBP recognized the underlying collagen structure and developed an aligned morphology (angle: 6.4 ± 4.8°),

whereas those on TCP displayed a polygonal shape and inconsistent alignment (angle: 34.6 ± 21.7°). Likewise, osteoblasts on BP, BD, and HP displayed random alignments. Osteoblasts adhered well on TCP, DBP, and BP, covering the entire surface, while they showed poor adhesion on BD and HP (Fig. 3B). Nucleus (DAPI) staining quantitatively confirmed a notably decreased osteoblast density on BD and HP (Fig. 3C). These results indicate that mineral-covered surfaces hinder cell adhesion and alignment along the collagen fibers, which is consistent with a previous report that found weaker cell adhesion on mineral surfaces[32].

We next characterized osteoblastic mineral deposition on DBP using fluorescent calcein. Confocal 3D microscopy revealed a distinct mineralization pattern between TCP and DBP. Osteoblasts on TCP formed small mineral nodules locally, whereas, on DBP, mineral deposition occurred on the entire surface, with small mineral granules gradually growing and developing into a fully mineralized layer (Fig. 3D, Supplementary Movie 1). Multiphoton microscopy confirmed that mineralization (green: calcein) occurred within collagen fibers (blue: second harmonic) on DBP (Fig. 3E). Time-course quantitative measurement of mineral deposition over 1 week confirmed that osteoblasts on DBP deposited significantly more mineral than osteoblasts on TCP (6-fold higher by Day 7) (Fig. 3F). Notably, the remineralized-DBP retained about 45% light transmittance, allowing microscopic accessing. The light transmittance of a bone disc (100 μm thick) was at 5.5%, which makes fluorescent imaging impractical (Fig. 3G). Finally, we conducted RNA sequencing of osteoblasts on TCP and DBP after 2 weeks of culture. The results showed a moderate difference in gene expression profiles, with fewer than 8% exhibiting notable fold change (fc) (1 <| Log2fc |) while genes related to ECM synthesis were expressed at higher levels on DBP (Supplementary Fig. 1). These results indicate that DBP serves as a functional template for the rapid and structural mineral deposition of osteoblasts, comparable to the mineralization of osteoid seen in bone tissue in vivo[33,34].

### Remineralized and decellularized DBP (RdBP) supports comparable osteoclastogenesis to bone discs

We hypothesized that remineralized and decellularized DBP (RdBP) facilitates osteoclastogenesis and mineral resorption comparable to BD, the gold standard for in vitro osteoclast assay, due to its preserved composition and structure of bone ECM. To test this, we cultured osteoblasts on DBP with an osteogenic differentiation medium for 1 week to remineralize DBP, then decellularized it with 0.1% SDS. Next, we coated the RdBP with calcein to allow fluorescent monitoring of the osteoclast mineral resorption process (Fig. 4A). We retrieved bone marrow from the femurs and tibias of eGFP mice and plated on TCP overnight to separate adherent stromal cells. We then introduced floating bone marrow monocytes (BMMs) to TCP, BD, and RdBP (1 ×10⁴ cells per mm²) and cultured them for 6 days with RANKL (40 ng/ml) and M-CSF (20 ng/ml) (Fig. 4B). To determine the role of bone ECM in directing osteoclastogenesis, we also included (i) collagen-coated TCP (col-TCP) with non-structural collagen, (ii) HP with non-structural mineral, (iii) DBP with structural collagen but without mineral, and (iv) thermally decomposed BD, referred as the mineral disc (MD), which has structural mineral but no collagen (Fig. 4C).

As expected, RANKL/M-CSF induced robust differentiation of BMMs into multinucleated osteoclasts, which was confirmed by tartrate-resistant acid phosphatase (TRAP) staining (Supplementary Fig. 2). Osteoclasts on col-TCP and HP showed comparable morphology to TCP, whereas those on DBP and MD showed similar morphology to BD (Fig. 4D). The size of osteoclasts on TCP-based substrates (TCP, col-TCP, HP) was 107.2 ± 101.4 μm². The significant size deviation was due to the continuous increase in the size of osteoclasts via repeated cell fusion until apoptotic death, which occurred when their size reached around 400 μm². The characterized size of osteoclasts on bone ECM-based substrates (DBP, BD, RdBP, MD) was 8.23 ± 0.42 μm²,

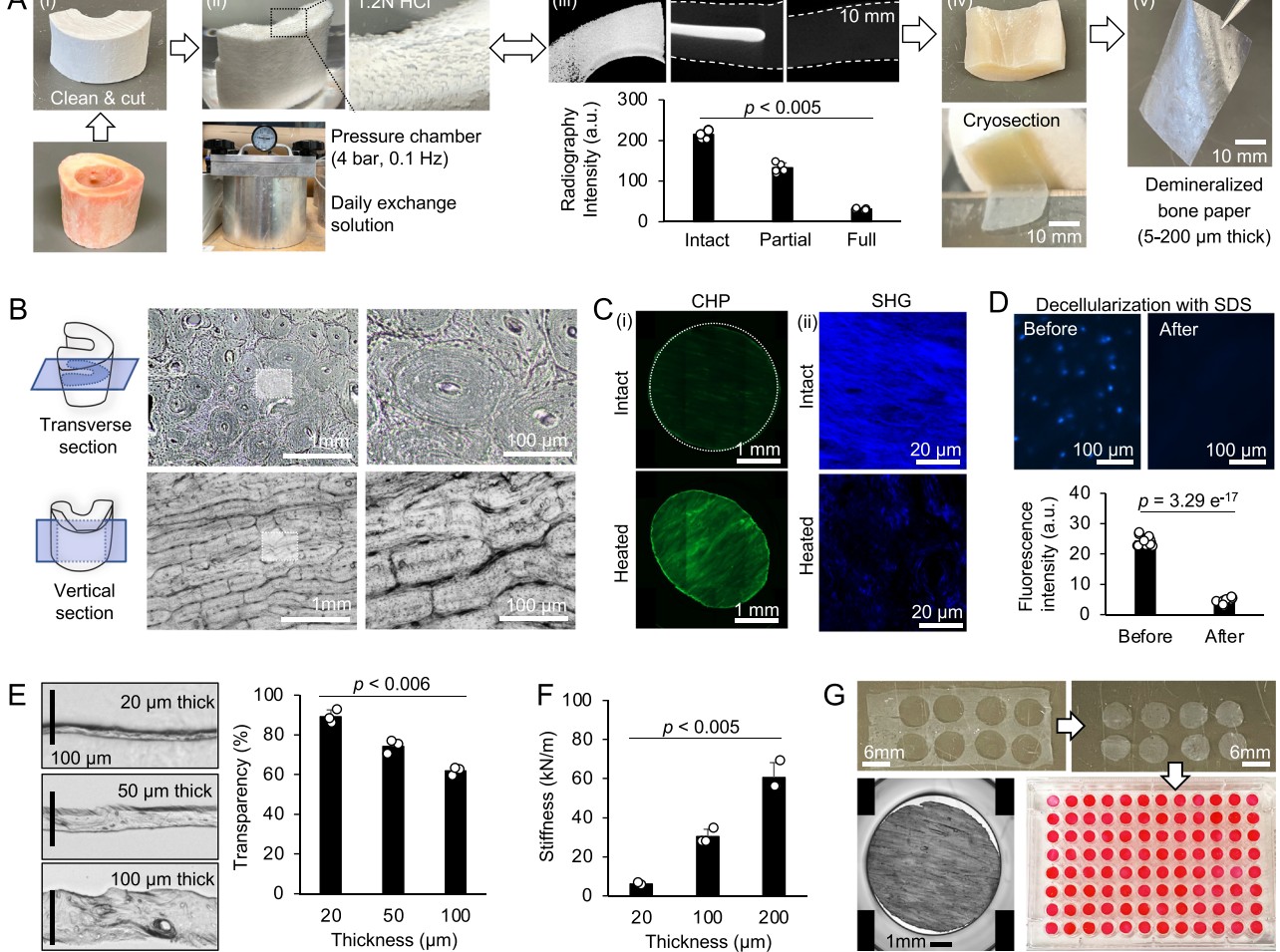

**Fig. 2 | Fabrication and characterization of demineralized bone paper (DBP).**
**A** (i) A bovine femur was cut into blocks and cleaned with a chloroform-methanol solution to dissolve residual fat. (ii) The compact bone block was demineralized in a 1.2 N of hydrochloric acid (HCl) solution under the cyclic pressure (0.1 Hz at 4 bar) for 5–7 days while daily replacing the solution. (iii) X-ray scanning was used to check for demineralization status. If residual minerals exist, the demineralization process is continued. (iv) The fully demineralized bone piece was cryo-sectioned. (v) A typical DBP is larger than 2 × 2 cm with a 20 μm thickness. **B** Representative microscopic images of DBP from routine sample preparations that show notably different collagen patterns depending on the cutting direction. **C** The biochemical integrity of collagen in DBP was confirmed by (i) fluorescent dye-conjugated collagen hybridizing peptides (CHP) that specifically bind to denatured collagen fibrils

and (ii) second harmonic generation (SHG) imaging that visualizes intact collagen fibrils under multiphoton microscopy. Heated DBP with denatured collagen was used as a positive control. Representative images from 3 independent experiments. **D** DBP was decellularized with 1% sodium dodecyl sulfate (SDS) and decellularization was confirmed by the nucleus (DAPI) staining ($n = 10$ independent samples). **E** Cross-sectional images of DBP with three different thicknesses and corresponding optical transparency based on an empty well ($n = 3$ independent experiments). **F** Characteristic mechanical stiffness of DBPs with three different thicknesses ($n = 3$ independent experiments). **G** Images of biopsy-punched DBP (D = 6 mm) placed in a 96-well plate. DBPs were stained with a red dye to visualize. The data are presented as mean ± standard deviation. $P$-values are derived from unpaired two-tailed $t$-tests. Related source data are provided as a Source Data file.

which is comparable to known osteoclast size in the body[14]. Osteoclasts on RdBP retain a comparable size during the extended RANKL/M-CSF stimulated culture for 2 weeks (Fig. 4E). The number of osteoclasts on TCP-based substrates was significantly less than bone ECM-based substrates due to frequent apoptosis (Supplementary Fig. 3). Time-lapse fluorescent imaging visualized the osteoclastic mineral resorption process on RdBP. Osteoclasts first made resorption pits, small and round resorption spots. As they continued to dissolve the mineral while migrating, a long and narrow trench pattern appeared (Fig. 4F, Supplementary Movie 2). These pit- and trench-type osteoclastic mineral resorption patterns on RdBP were similar to those previously reported on BD[35,36]. One notable observation was that osteoclasts on TCP-based substrates exhibited a slender actin belt along their edges, while osteoclasts on bone ECM-based substrates formed a comparatively thick actin ring, positioned a considerable distance from their periphery (Fig. 4G). These results indicate that preserving the structural intricacies of bone ECM, whether mineral or

collagen, is crucial for recapitulating osteoclast processes via directing actin ring formation[37–39].

Next, we evaluated the analytical potential of RdBP compared to HA, which is commonly used to quantitatively monitor osteoclastic mineral resorption by leveraging the reduced transmission contrast. On RdBP, mineral resorption was tracked by observing the local area with decreased fluorescent intensity (Fig. 4H, Supplementary Movie 3). Quantitative characterization of mineral resorption kinetics revealed that osteoclasts on the HP dissolved over 80% of the mineral surface within 24 h. Osteoclasts on RdBP removed about 30% of the mineral surface by 48 h, after which mineral resorption stopped (Fig. 4I). This may be because most osteoclasts underwent apoptosis that rapidly exhausts precursor BMMs after 48 h of bone resorption. The same experiment on BD was limited for endpoint mineral resorption characterization due to its opaqueness. On BP, mineral resorption was evident by reducing bone particle sizes, but the co-existence of TCP and bone particles restricted reproducible quantitative analysis

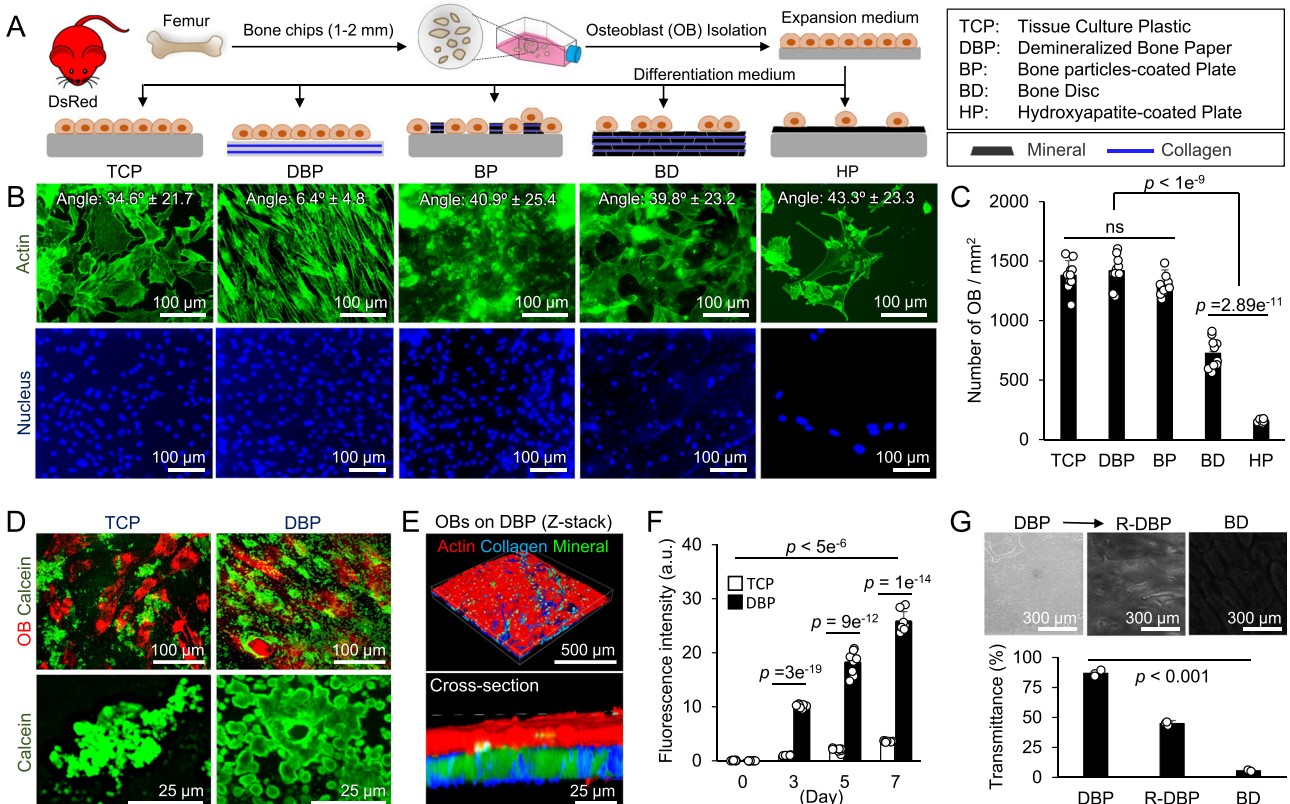

**Fig. 3 | Comparison of osteoblast adhesion and mineralization on existing osteoclast assay substrates. A** Primary osteoblasts from DsRed reporter mice were harvested from femoral bone chips, expanded on TCP, and cultured on 5 different substrates. The red mouse image has been obtained from Clker.com and flyclipart.com. **B** Representative fluorescent microscopy images of the nucleus (blue: DAPI) and actin (green: Phalloidin) after 1-week of culture and measured cell alignment angles on different substrates ($n = 40$ independent samples). **C** A comparison of OB growth after 1 week of culture on different substrates ($n = 10$ independent samples). **D** Fluorescent microscopy images of osteoblasts (DsRed) and mineral (green: calcein) show rapid and homogenous mineral deposition on DBP

compared to TCP. **E** 3D multiphoton microscopy images of remineralized-DBP by osteoblasts after 1-week of culture (red: actin-phalloidin, green: mineral-calcein, blue: collagen-second harmonic) show mineral deposition in DBP collagen. **F** Time-course measurement of mineral deposition with calcein staining for 7 days on TCP and DBP ($n = 8$ independent samples). **G** Transmission microscopy images of DBP, remineralized-DBP (R-DBP), and bone discs (BD), and quantitative comparison of their transmittance ($n = 3$ independent experiments). The data are presented as mean ± standard deviation. P-values are derived from unpaired two-tailed t-tests. ns not significant ($p > 0.05$). Related source data are provided as a Source Data file.

(Supplementary Fig. 4). Overall, these results demonstrate the enabling features of RdBP, which support an accurate recapitulation of in vivo-relevant osteoclastogenesis while continuously monitoring the processes.

**Biochemical stimulation of osteoblasts and BMMs coculture on DBP recapitulates a bone remodeling cycle**

Osteoblast-osteoclast coculture is essential for accurately replicating osteoclast biology[40]. We hypothesized that osteoblasts on DBP regulate osteoclast differentiation of BMMs by altering their secretion profile of regulatory molecules, depending on their metabolic state, either resting or activated by vitamin D3 (VD3) and prostaglandin E2 (PGE2) stimulation, instead of directly adding RANKL and M-CSF (Fig. 5A). After culturing osteoblasts on DBP, TCP, and BD for 1 week, we measured OPG and RANKL secretion using ELISA. Osteoblasts on DBP and TCP secreted relatively high levels of OPG and low levels of RANKL, indicating an inhibitory molecular milieu for osteoclastogenesis at a resting state. In contrast, osteoblasts on BD secreted similar levels of OPG and RANKL. Following VD3/PGE2 stimulation, the secretion profile of OPG and RANKL shifted, with the most pronounced change observed on DBP (Fig. 5B).

We also performed RNA sequencing analysis to monitor the changes in global gene expression patterns of osteoblasts on DBP when transitioning from a resting state to a VD3/PGE2 stimulated state. In the total of 17,258 genes analyzed, approximately 5,000 showed

statistically significant expression (Q-value < 0.01), and about 20% of these genes exhibited notable fold change (fc) ($1 <| \text{Log2fc} |$) (Fig. 5C, Supplementary Data 1). We then focused on 128 selected genes related to bone metabolism and visualized their up- and downregulation following VD3/PGE2 stimulation via a volcano plot (Fig. 5D). We listed the profile of individual genes under functional categories. As expected, *Vdr* (vitamin D receptor) and *Tnfrsf11* (RANKL) were upregulated, and *Tnfrsf11b* (OPG) was downregulated, while *Tnfrsf11b* (RANK receptor) showed no significant change. A subset of genes associated with bone formation (*Dmp1, Bglap, Phex, Col1a2, Col2a1, Runx2, Bmp4, Alpl*) were downregulated, while monocyte chemotactic proteins (*Ccl2, Ccl6, Ccl8, Ccl12, Cxcl12*), growth factors (*CSF3, FGF2*), inflammatory molecules (*IL1b, IL6, Ptgs2*), and matrix metalloproteinases (*Mmp8, Mmp10*) were upregulated (Fig. 5E). These results indicate that VD3/PGE2 stimulation reduces bone formation while promoting BMM attraction, proliferation, differentiation into osteoclasts, and subsequent bone resorption, which is consistent with previously reported osteoporosis-associated genes[41].

Next, we recapitulated a bone remodeling cycle by coculturing eGFP osteoblasts and DsRed BMMs on DBP (Fig. 6A). Increased RANKL secretion by VD3/PGE2 stimulated osteoblasts induced BMMs to differentiate into multinucleate osteoclasts, as confirmed by time-course fluorescent images of osteoblasts (red) and BMMs (green) (Supplementary Movie 4). On TCP, osteoclasts exhibited abnormally large sizes as BMMs fused repeatedly until apoptosis. When VD3/PGE2 were

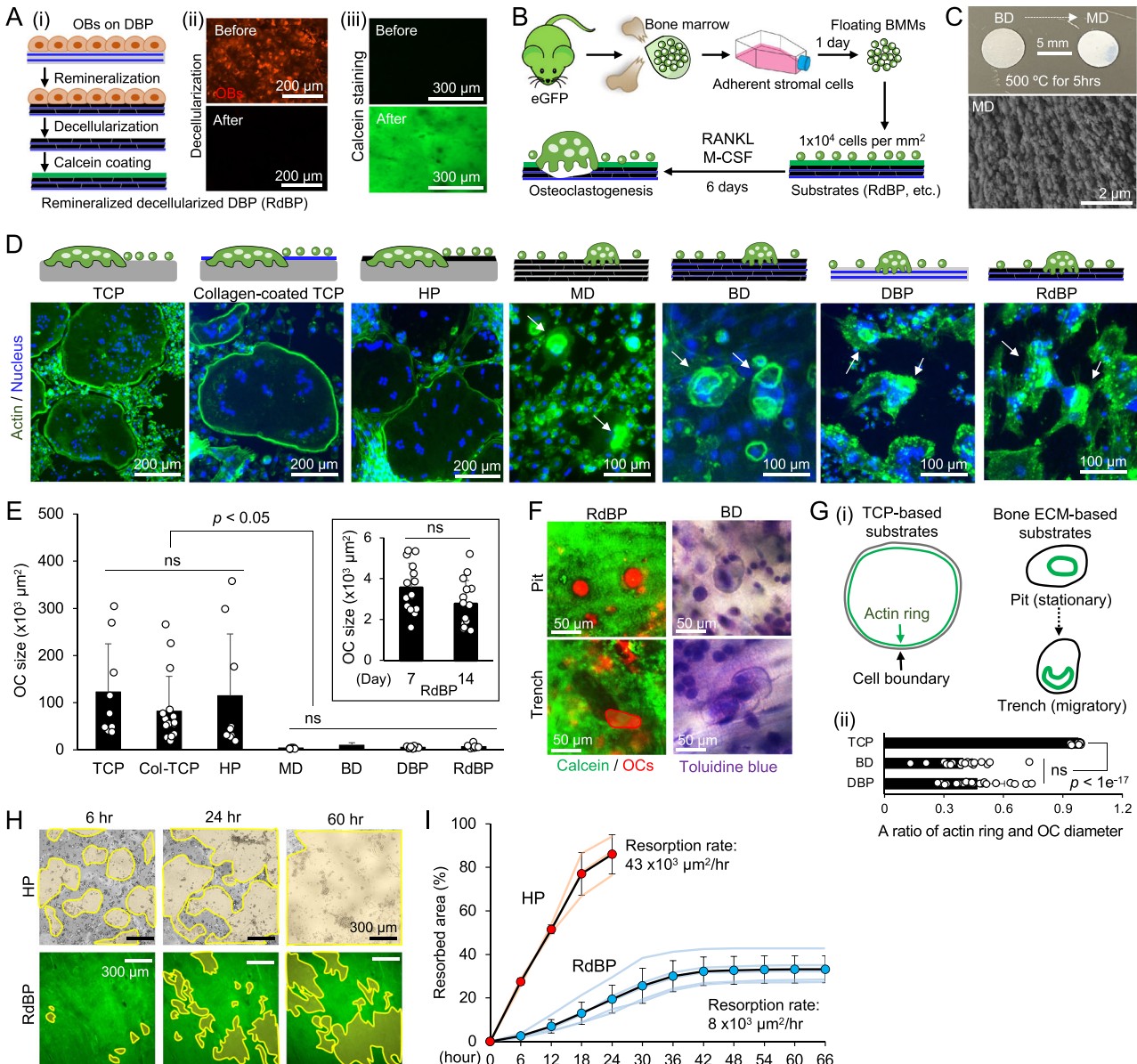

**Fig. 4 | Comparative osteoclast culture and assay on remineralized and decellularized DBP (RdBP) with existing platforms. A** (i) Schematic of RdBP preparation. A green layer indicates calcein coated on RdBP. (ii) Confirmation of decellularization after osteoblast remineralization of DBP with SDS. (iii) Fluorescent calcein-coated RdBP. Representative images from 5 independent experiments. **B** Schematic of osteoclastogenesis assay by adding RANKL/M-CSF in murine BMM culture. The green mouse image has been obtained from Clker.com and flyclipart.com. **C** Preparation of mineral disc (MD) by thermal decomposition of BD, with its surface morphology under scanning electron microscopy imaging. Representative images from 3 independent experiments. **D** Representative fluorescent images of mature osteoclasts (green: phalloidin, blue: DAPI) on seven different substrates (Col-TCP: collagen-coated TCP). **E** Quantified osteoclast size after 7 days of RANKL/M-CSF stimulated culture on different substrates. (TCP, HP, MD, BD, RdBP: $n = 9$, Col-TCP: $n = 16$, DBP: $n = 14$ independent samples). The inner panel shows osteoclast size after 14 days of stimulated culture on RdBP. ($n = 15$ independent samples) **F** Comparison of pit and trench-type bone resorption patterns between RdBP and BD. **G** (i) Schematic of generalized osteoclast actin ring structure on TCP-based substrates (TCP, Col-TCP, HP) and bone ECM-based substrates (MD, BD, DBP, RdBP). (ii) Quantitative comparison of a ratio of actin ring and osteoclast boundary ($n = 20$ cells from 3 independent experiments). **H** Time-lapse microscopic monitoring of osteoclastic mineral resorption on HP and RdBP. Resorbed mineral areas are contoured with yellow lines. **I** Comparative time-course measurements of osteoclastic mineral resorption area on HP and RdBP with bone resorption rates (HP: $n = 3$, RdBP: $n = 5$ independent samples). The data are presented as mean ± standard deviation. P-values are derived from unpaired two-tailed t-tests. ns: not significant ($p > 0.05$). Related source data are provided as a Source Data file.

withdrawn, the size and number of osteoclasts gradually decreased on both TCP and DBP (Fig. 6B, Supplementary Movie 5). Time-course fluorescent imaging enabled quantitative tracking of fusion, fission, and apoptosis processes in eGFP-BMMs (Fig. 6C, Supplementary Movie 6 & 7). About 96 h after VD3/PGE2 stimulation, mature osteoclasts (>3 nuclei) began to emerge on DBP. Between 72 and 120 h, BMM fusion was the primary event. From 120 h, osteoclasts maintain a

similar size with decreased fusion but increased fission. Following the cessation of VD3/PGE2 (144–196 h), osteoclasts began to reduce in size due to increased fission and eventually reverted to monocytes. During this period, a subset of osteoclasts underwent apoptosis (Fig. 6D). On TCP, osteoclast fusion began about 72 h after VD3/PGE2 stimulation, and the rate of fusion decreased after 120 h of stimulation, comparable to DBP. However, the rate of fusion was approximately 4 times higher

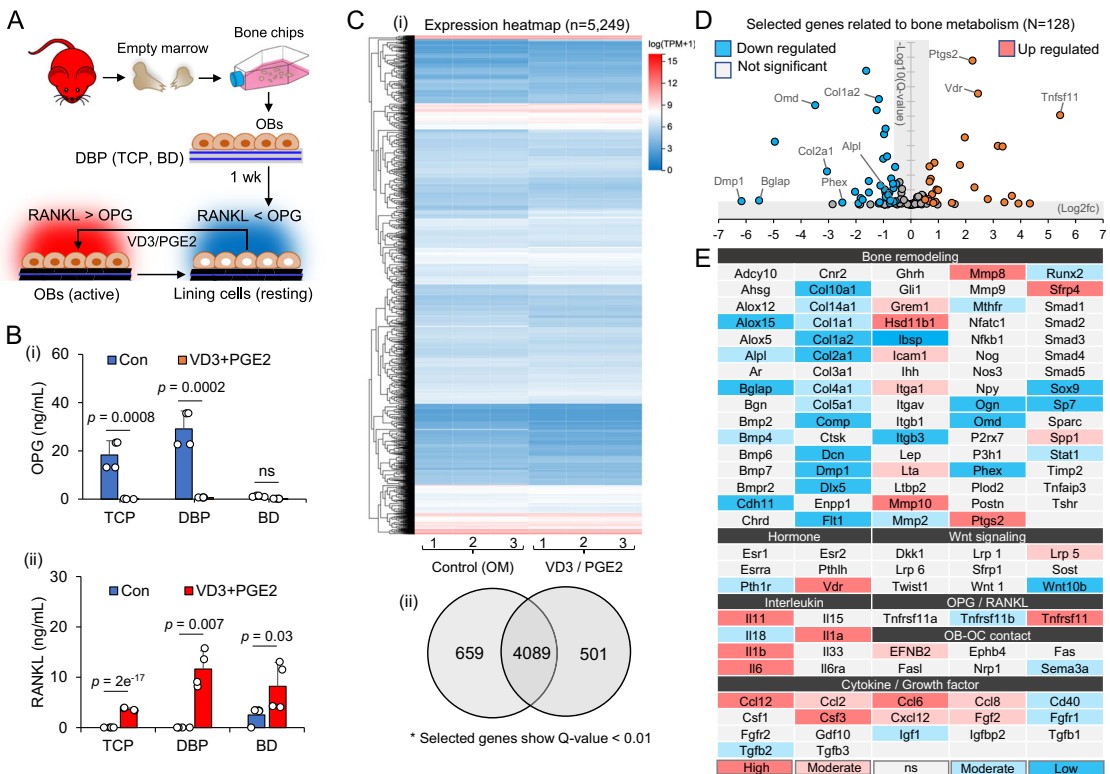

**Fig. 5 | Comparative characterization of osteoblast secretion and gene expression profiles in DBP between the resting and stimulated states.**
**A** Schematic of manipulating osteoblast metabolic state; Osteoblasts cultured on DBP for 1 week acquire resting state lining cell phenotype. When stimulated with vitamin D3 (VD3) and prostaglandin E2 (PGE2), lining cells revert to active osteoblasts. The red mouse image has been obtained from Clker.com and flyclipart.com. **B** Comparative RANKL and OPG secretion profiles of osteoblasts on TCP, DBP, and BD (i) before and (ii) after VD3/PGE2 stimulation ($n = 4$ independent samples). The data are presented as mean ± standard deviation. $P$-values are derived from unpaired two-tailed $t$-tests.

ns not significant ($p > 0.05$). Related source data are provided as a Source Data file. **C** (i) Comparative RNA-seq heat-map of osteoblasts on DBP between resting and VD3/PGE2 stimulated states for selected genes having the Q-value lower than 0.01 (total: 5,249). (ii) Distinct and overlapped gene numbers between resting and activated osteogenic cells. **D** A volcano plot of 128 selected bone remodeling-related genes (up-regulated: Log2fc > 0.5, down-regulated: Log2fc < −0.5). **E** List of 128 genes and their profiles of up- and down-regulation, and non-significant changes (different colors to divide them further into high or low: 1 <| Log2fc |, moderate: 0.5 <| Log2fc | <1). Original RNA-seq results are provided as a Supplementary Data.

than DBP. Upon withdrawal of VD3/PGE2 (144–196 h), fusion occurred for more than 24 h, which was distinct from DBP where fusion stopped within 6 h of cessation of VD3/PGE2. A significantly less fission and relatively low apoptosis occurred on TCP during the stimulation. Upon withdrawal of VD3/PGE2, apoptosis rates on TCP increased about 1.5 times compared to DBP. However, given the notably larger osteoclast morphology on TCP, a significantly higher frequency of apoptosis on TCP per BMMs was anticipated (Fig. 6E). These results substantiate differences and similarities of osteoclastogenesis between DBP and TCP, while demonstrating the analytical power of DBP-based osteogenic cell culture.

We then investigated the role of osteoblasts in maintaining the survival and function of BMMs and osteoclasts by comparing osteoclastogenesis between BMM single culture on RdBP with RANKL/M-CSF and BMM and osteoblast coculture on DBP with VD3/PGE2. We quantitatively visualized osteoclastic mineral resorption by staining minerals with fluorescent calcein. In BMM single cultures, we observed significant (~35%) mineral resorption within 2 days, but there was no further resorption after this. However, mineral resorption in BMM and osteoblast cocultures began slowly but steadily continued by the end of the experiment (Supplementary Movie 8). The mineral resorption rate in single cultures ($8 \times 10^3$ µm²/hour) was 4 times higher than that in cocultures ($2 \times 10^3$ µm²/hour) (Fig. 6F). These findings provide strong evidence that osteoblasts play a major role in maintaining the proliferation and differentiation of BMMs via secreted factors and possibly by direct cellular contact[16,29,42,43]. Overall, our study demonstrates the potential of DBP-based coculture as a powerful model for

recapitulating osteoclast-related biological processes with high fidelity and analytical power.

**Osteoblasts and BMMs coculture on DBP recapitulates functional bisphosphonate drug responses**
We examined whether the osteoblast and BMM coculture on DBP could functionally and analytically recapitulate osteoclast-targeting bisphosphonate drug action. We used osteosense680 (OS680), a bisphosphonate-conjugated fluorescent dye, to visualize mineral binding and osteoclast targeting function. After 1 week of osteoblast culture on DBP, we introduced OS680 (5 µM) for 24 h and then washed with PBS to simulate bone mineral binding of systemically delivered bisphosphonate drugs. Fluorescent microscopy confirmed the homogenous attachment of OS680 on remineralized DBP (Fig. 7A). Next, we examined the osteoclast-targeting function of bisphosphonate by coculturing osteoblasts and BMMs on DBP with and without OS680 coating under VD3/PGE2 stimulation for 6 days (Fig. 7B). Confocal microscopy confirmed that osteoclasts differentiated on OS680-coated DBP reduced the local fluorescent intensity (Fig. 7C). Time-laps fluorescent images (green) captured osteoclasts emerged on OS680-coated DBP quickly underwent apoptosis (Supplementary Movie 9). Caspase-3/7 dye confirmed the osteoclast apoptosis on OS680-coated DBP (Fig. 7D, Supplementary Movie 10). The characteristic lifespan (from the completion of osteoclast fusion to fission or apoptosis) of osteoclasts in OS680-coated DBP (39 ± 20 h) was considerably shorter than control DBP (134 ± 42 h). The number of apoptotic cells confirmed by Casepase-3/7 dye was 3-fold higher in the

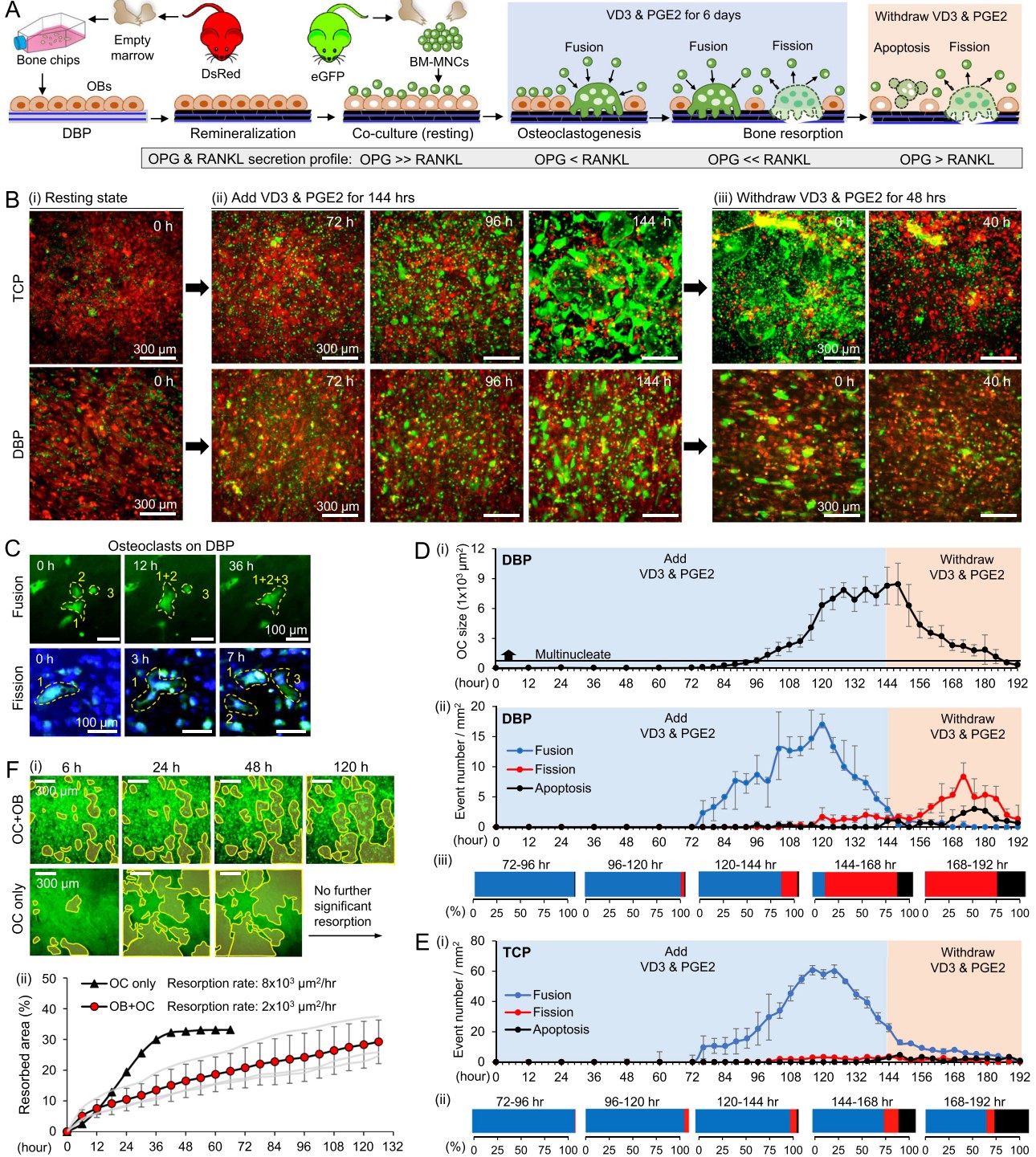

**Fig. 6 | Coculture osteoblasts and BMMs on DBP recapitulates a biochemical stimulation-induced bone remodeling cycle. A** Schematic of experimental strategy to recapitulate and monitor a bone remodeling cycle and mineral resorption by stimulating osteoblasts on DBP cocultured with BMMs with VD3/PGE2. The red and green mouse images have been obtained from Clker.com and flyclipart.com. **B** Representative time-course images of osteoblast (red) and BMMs (green) coculture on TCP and DBP (i) resting, (ii) during VD3/PGE2 stimulation, and (iii) withdrawing VD3/PGE2. **C** Representative images of osteoclast fusion and fission (blue: DAPI). **D** (i) Time-course measurement of OC sizes during a VD3/PGE2 stimulation-induced bone remodeling cycle of osteoblasts and BMMs coculture on DBP (n = 10 independent samples). (ii) Time-course quantified cellular events of osteoclast fusion, fission, and apoptosis (n = 3 independent samples) and (iii)

comparative cellular event ratios (n = 3 independent samples) during active remodeling periods on DBP. Representative time-course data from three independent experiments. (blue: fusion, red: fission, black: apoptosis) **E** (i) Time-course quantified cellular events of osteoclast fusion, fission, and apoptosis and (ii) comparative cellular event ratios during active remodeling periods on TCP (n = 3 independent samples). Representative time-course data from three independent experiments. **F** (i) Representative comparative time-course images of mineral resorption areas between osteoblast-osteoclast coculture on DBP and osteoclast single culture on RdBP. (ii) Time-course quantification of resorption areas (n = 3 independent samples). The data are presented as mean ± standard deviation. Related source data are provided as a Source Data file.

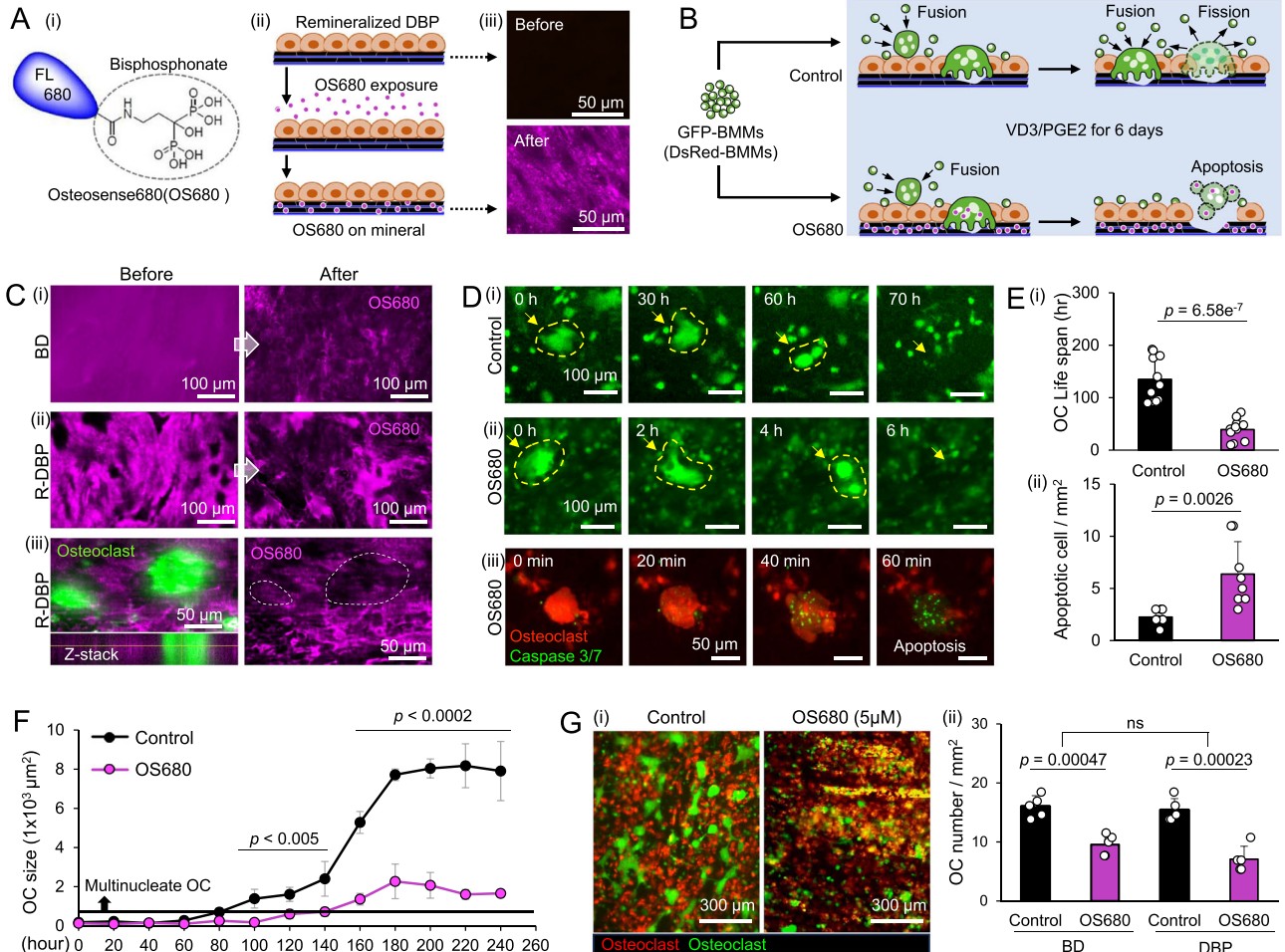

**Fig. 7 | Ostensense680 (OS680) and fluorescent BMMs recapitulate bisphosphonate responses to osteoclasts in osteoblasts and BMMs coculture on DBP under VD3/PGE2 stimulation. A** (i) OS680: fluorescent dye-conjugated bisphosphonate, (ii) experimental procedure to simulate bisphosphonate drug treatment, and (iii) confirmation of OS680 adhesion on re-mineralized DBP (R-DBP) under fluorescent microscopy. **B** Schematic of experiments to simulate bisphosphonate drug action on osteoblast and BMM coculture on DBP. **C** 3D confocal microscope images show decreased OS680 fluorescent intensity after osteoclast resorption on (i) BD and (ii) R-DBP. (iii) Z-stack image confirms locally decreased OS680 fluorescent intensity where osteoclasts reside. **D** Representative time-lapse fluorescent images of (i) osteoclast fission on control coculture, (ii) Osteoclast apoptosis on OS680-coated coculture, and (iii) confirmation of osteoclast apoptosis using caspase 3/7 fluorescent dye. **E** Characterized (i) life span of osteoclasts with and without OS680 treatment ($n = 12$ independent samples) and (ii) normalized number of apoptotic osteoclasts ($n = 7$ independent samples). **F** Time-course measurement of osteoclast size increase during osteoblasts and BMMs coculture on DBP with or without OS680 treatment ($n = 5$ independent samples). **G** (i) Representative fluorescent images of coculture DBPs in control and OS680 after 10 days of VD3/PGE2 exposure (green: BMMs, Osteoclasts, red: Osteoblasts). (ii) Quantitative comparison of osteoclast number between control and OS680-treated coculture on DBP and BD ($n = 5$ independent samples). The data are presented as mean ± standard deviation. *P*-values are derived from unpaired two-tailed *t*-tests. ns not significant ($p > 0.05$). Related source data are provided as a Source Data file.

OS680-coated DBP (Fig. 7E). These results indicate that osteoclasts degrade OS680-attached remineralized DBP and subsequently uptake the released bisphosphonate, possibly via pinocytosis, which eventually induces their apoptosis[44].

Time-course measurements of osteoclast size and number with and without OS680 treatment demonstrated how bisphosphonate regulates osteoclast differentiation. Multinucleated osteoclasts larger than $1 \times 10^3$ μm² emerged ~80 h after VD3/PGE2 treatment in the control group, but ~140 h in the OS680 treatment group. As shown before, the size of osteoclasts stabilized around $8 \times 10^3$ μm². Notably, the size of osteoclasts in the OS680 treatment group remained around $2 \times 10^3$ μm² as mature osteoclasts underwent apoptotic death (Fig. 7F). The significantly decreased osteoclast density on OS680-treated DBP at the end of the experiment (10 days) was comparable to the counter experiment on OS680-treated BD (Fig. 7G). These results support that the potential of the osteoblast and osteoclast coculture model on DBP to faithfully recapitulate osteoclast-targeting drug processes in the body.

## Humanized RdBP facilitates quantitative assessment of bisphosphonate drug responses in human osteoclasts

To evaluate the applicability of established murine osteoclast culture and assay platforms to human osteoclasts, we first established human bone marrow stromal cells by isolating and expanding adherent cells from healthy donors' fresh bone marrow aspirates. We seeded these cells onto DBP and TCP and cultured them for 2 weeks in an osteogenic differentiation medium to differentiate into osteoblasts (Fig. 8A). As expected, human osteoblasts adhered to DBP following the underlying collagen fibers and rapidly deposited minerals when compared to TCP (Fig. 8B). We then prepared a humanized RdBP by decellularization of remineralized DBP and subsequent coating with fluorescent calcein. Next, we isolated human CD14[+] monocytes from peripheral blood and cultured them for 1 week with human M-CSF (hM-CSF) (20 ng/ml) to expand their number. We then seeded hCD14[+] cells onto TCP, HP, and RdBP ($1 \times 10^4$ cells per mm²) and cultured them for 2 weeks in the presence of hM-CSF and human RANKL (hRANKL) (40 ng/ml) (Fig. 8C). On TCP

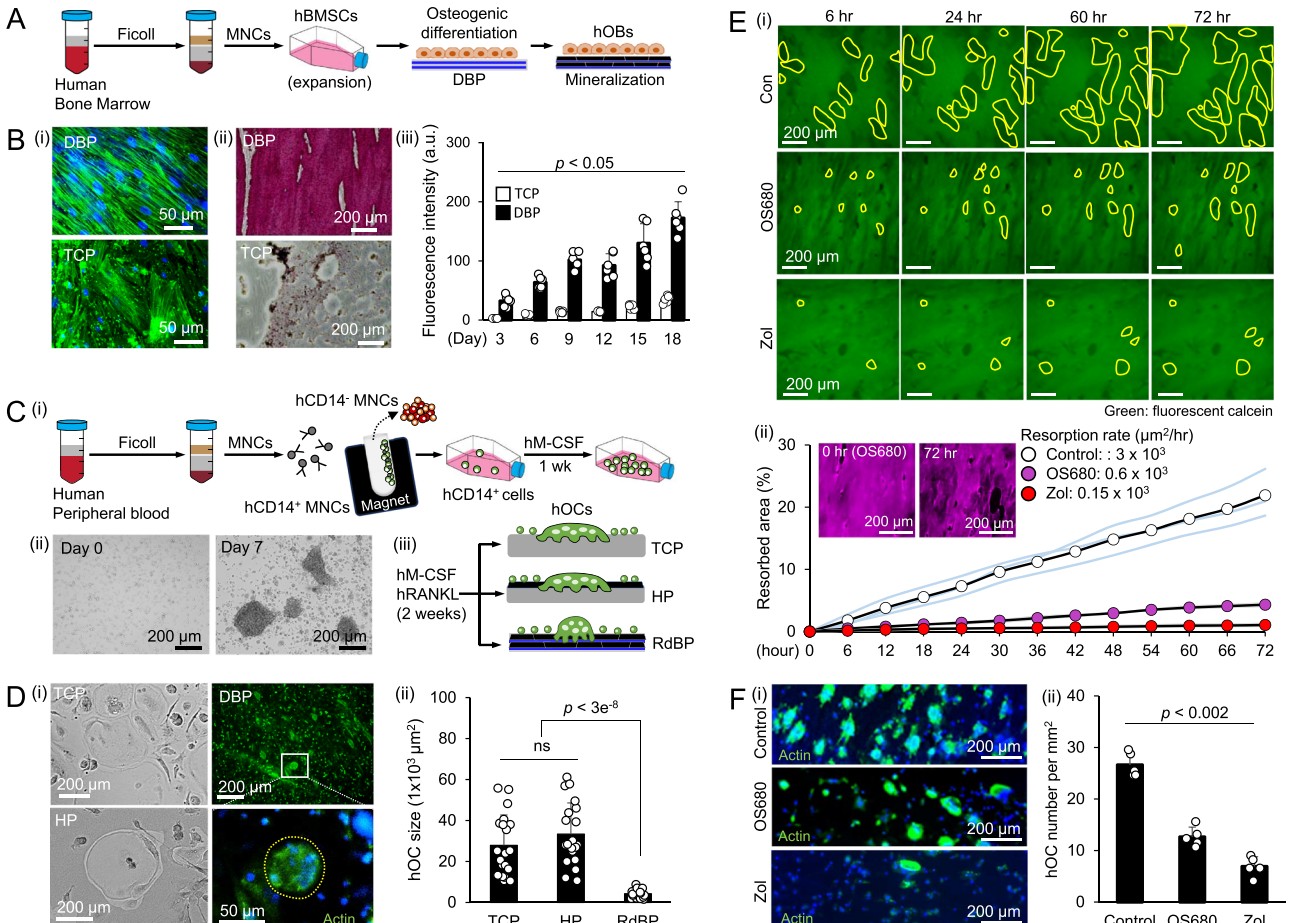

**Fig. 8 | Human osteoclasts (hOCs) response to bisphosphonate drugs on humanized RdBP. A** Schematic of human bone marrow stromal cells (hBMSCs) isolation, expansion, and differentiation into human osteoblasts (hOBs) on DBP. After 2 weeks of remineralization in osteogenic differentiation medium, humanized RdBP was prepared by decellularization. **B** (i) hOBs adhere following the collagen fiber orientation on DBP but randomly on TCP. (ii) Alizarin red mineral staining after hOB culture on DBP and TCP. (iii) Time-course quantitative measurement of mineral deposition by fluorescent calcein staining ($n = 6$ independent samples). **C** (i) Schematic of human CD14+ mononuclear cells (hCD14+ MNCs) isolation and expansion from peripheral blood using magnetic-activated cell sorting system. (ii) Brightfield images of ex vivo expanded hCD14+ cells for 1 week culture with hM-CSF. Representative images from 3 independent experiments. (iii) hCD14+ cells cultured on TCP, HP, and RdBP for 2 weeks with hM-CSF and hRANKL to induce hOC differentiation. **D** (i) Representative images of hOCs on TCP, HP, and RdBP. (green: phalloidin, blue: DAPI) (ii) Characterization of hOC sizes on TCP, HP, and RdBP after 2 weeks of the stimulated culture ($n = 20$ cells from three independent experiments). **E** (i) Representative time-course fluorescent images of calcein-coated mineral resorption by hOCs on control, OS680-, and Zoledronate (Zol)-treated RdBP. (ii) Time-course quantitative monitoring of resorption areas. ($n = 3$ independent samples) Inner panel images show mineral resorption (purple: OS680). **F** (i) Representative fluorescent images of hOCs on control, OS680-, and Zol-treated RdBP (green: phalloidin, blue: DAPI). (ii) Normalized hOC density at the end of 2 weeks culture ($n = 5$ independent samples). The data are presented as mean ± standard deviation. *P*-values are derived from unpaired two-tailed *t*-tests. ns: not significant ($p > 0.05$). Related source data are provided as a Source Data file.

and HP, human osteoclasts displayed a spread morphology with significant size variation, whereas on humanized RdBP, osteoclasts showed a smaller and rounded morphology (Fig. 8D).

Finally, we assessed the capability of humanized RdBP as a preclinical screening platform for testing bone-targeting osteoporosis drugs. We treated humanized RdBP with OS680 (5 μM) and zoledronate (5 μM). OS680 is based on pamidronate[45], a less potent bisphosphonate drug than zoledronate[46]. As expected, time-course quantitative imaging analysis substantiated that zoledronate-treated RdBP showed a lesser reduction in mineral resorption compared to OS680-treated RdBP (Fig. 8E). At the end of the experiment, we fixed the culture and conducted actin and nucleus staining to visualize multinucleated osteoclasts. The results showed a significantly decreased number of human osteoclasts on zoledronate-treated RdBP than on OS680-treated RdBP (Fig. 8F). These results demonstrate that humanized RdBP can be used to predict the clinical outcomes of osteoclast-targeting osteoporosis drugs.

## Discussion

Biomaterial substrates for in vitro osteoclast assays should meet at least three critical criteria. First, the biomaterial should support the differentiation and function of osteoclasts in a manner that is similar to what is observed in vivo, including cell morphology, lifespan, and resorption activity. Second, the biomaterial should be transparent or translucent enough to allow direct microscopic observation of osteoclasts for quantitative analysis of osteoclastogenesis and bone resorption. Third, the biomaterial should be easy to produce and handle, and it should yield consistent results from experiment to experiment. Unfortunately, most existing assay platforms do not fully meet all these criteria. TCP and HP are transparent and standardized, which allows direct microscopic observation of osteoclastogenesis and reproducible experiments. However, osteoclasts on TCP and HP exhibit abnormally large morphology and short lifespan, which does not match in vivo observations and could mislead in vitro findings. BD preserving natural bone ECM supports in vivo-relevant osteoclast phenotype and resorption, making them the gold standard for in vitro

osteoclast assays. However, their practical and broad application has been challenging due to burdensome preparation, opaqueness for microscopic imaging, and suboptimal adhesion and growth of osteoblasts with pre-existing minerals. BP has shown potential for quantitative monitoring of osteoclastic bone resorption. However, the irregular size and shape of bone particles and co-development of osteoclasts on both BP and TCP can restrict standardized and reproducible assay developments (Fig. 1).

We have developed an osteoclast assay platform based on osteoid-mimicking DBP, which demonstrated significant advantages over existing platforms (Fig. 2). First, remineralized-DBP by osteoblasts exhibits a bone-like ECM structure and composition, which enables in vivo-relevant osteoclast differentiation and function. The importance of bone ECM in reproducing osteoclast morphology and mineral resorption has been reported[47,48]. Second, remineralized-DBP retains sufficient optical transparency for direct fluorescent imaging, which allows time-course quantitative monitoring of multicellular processes, including osteoclast fusion, fission, apoptosis, and bone resorption, which remains challenging on BD. Essentially, remineralized-DBP is a 20 μm thick bone, 5–25 times thinner than BD (100–500 μm thick). Although reducing the thickness of BD is doable, the resultant 20 μm thick BD becomes too brittle to handle (Fig. 3). Third, DBP is scalable for production and mechanically durable, which significantly facilitates the handling of DBP and running scalable experiments. One bovine femur provides enough DBPs to prepare more than five hundred 96-well plates layered with DBP. These features enable faithful and scalable recapitulation of in vivo-relevant osteoclast differentiation, mineral resorption, and time-course quantitative monitoring of multicellular processes, required for high-content and high-throughput assay development.

We demonstrated compelling evidence that our in vitro bone model on DBP recapitulates critical bone remodeling-related biological processes via time-course quantitative fluorescence microscopic imaging (Fig. 6). First, osteoclasts differentiated on remineralized DBP are comparable in size and mineral resorption patterns to those observed on BD and in vivo studies. Second, the co-culture of osteoblasts and BMMs on DBP under VD3/PGE2 stimulation supports osteoclastogenesis with increased secretion of RANKL by stimulated osteoblasts, rather than directly adding RANKL. Third, introducing and withdrawing VD3/PGE2 in the co-culture model direct a bone remodeling cycle, which allows robust reproduction of osteoclast fusion and fission. The size of osteoclasts becomes stabilized, suggesting a balanced osteoclast fusion and fission events. Lastly, osteoblasts maintain the pool of BMMs and support their survival, proliferation, and differentiation on DBP. These enabling features of our in vitro bone model based on DBP are expected to contribute greatly to understanding the mechanisms behind many enigmatic bone remodeling processes and facilitate the identification of target drug compounds for bone metabolism.

We compared osteoclastogenesis on systematically modified bone ECM (BD, DBP, MD) and synthetic substrates (TCP, Col-TCP, HP) to elucidate the role of bone components and structure in regulating osteoclast differentiation (Fig. 4). Our findings suggest that minerals are not essential for osteoclastogenesis, but retaining the bone ECM is critical to restoring in vivo-relevant osteoclast phenotype and function. The distance between the actin ring and the osteoclast periphery may be a key determinant of osteoclast fusion. On synthetic substrates, osteoclasts exhibit a belt-like thin actin structure positioned close to the cell periphery, which could facilitate fusion upon contact with BMMs and osteoclasts. Conversely, on bone ECM substrates, osteoclasts exhibit a smaller, denser actin ring located approximately 5–20 μm inner side from the cell periphery, which could reduce osteoclast fusion and potentially apoptosis. The hierarchical ultrastructure of bone ECM may play a central role in developing the actin ring and its position within an osteoclast. Overall, our findings warrant

further detailed mechanistic investigation of the role of bone ECM in regulating osteoclastogenesis.

Inducing a resting state in osteoblasts by culturing them on DBP enabled the recapitulation of a bone remodeling cycle because subsequent VD3/PGE2 stimulation significantly shifted OPG and RANKL secretion, which triggers osteoclast differentiation of co-cultured BMMs. RNA sequencing data substantiated the shifted genes from resting to activated states of osteoblasts. During VD3/PGE2 stimulation, osteogenesis-related genes, such as those involved in collagen synthesis and mineral deposition, are downregulated, whereas genes involved in attracting BMMs and supporting their growth and differentiation are upregulated (Fig. 5). This explains the prolonged maintenance of BMMs on DBP-based bone models. One caveat is that RNA sequencing was only performed at resting and activated states, but not in a transitioning state. Previous studies have shown that continuous parathyroid hormone treatment negatively impacts bone mass, but intermittent treatment improves bone mass[49]. Similarly, withdrawing VD3/PGE2 may upregulate osteogenesis-related genes while downregulating osteoclastogenesis genes. Further timecourse RNA sequencing analysis would provide a more comprehensive understanding of this process.

In our proof-of-concept study, we demonstrated the potential of the DBP-based bone model to simulate a clinical scenario of bisphosphonate treatment with OS680. This fluorescent dye-conjugated bisphosphonate visualizes its attachment to remineralized DBP and subsequent release during bone resorption by mature osteoclasts. We monitored the ensuing apoptosis of these active osteoclasts that uptake OS680 using caspase-3/7 live-cell dye (Fig. 7). To rapidly analyze osteoclast apoptosis, we developed an algorithm that leverages the automatic cell sorting function in CellProfiler (Supplementary Fig. 5). Visualizing bisphosphonate action against osteoclasts is important because it could deepen our understanding of bonetargeting osteoporosis medications. An earlier study on BD used fluorescent bisphosphonate to illustrate mineral binding and the uptake of bisphosphonate by osteoclasts by fixed and antibodystained cell culture under 3D confocal imaging[50]. The DBP-based bone model is poised to facilitate the visualization of bone remodeling and drug responses, tapping into existing functional molecular probes. This is a significant advance, as it could enable researchers to study the effects of bisphosphonates and other drugs on bone remodeling in a more realistic and controlled in vitro setting.

Our DBP-based bone model is applicable to the mechanistic investigation of other osteoporosis drugs, which can be categorized into three groups: (i) bisphosphonate-based drugs, such as alendronate and zoledronate, that bind to bone minerals and induce apoptosis of osteoclasts during bone resorption[51,52]; (ii) monoclonal antibody-based drugs, including denosumab and romosozumab, that modulate receptor-ligand signaling for osteoclast differentiation;[53,54] and (iii) hormonal replacement drugs, like parathyroid hormone and estrogen, that promote an anabolic effect on bone-forming osteoblasts[55,56]. Long-term prescription of these drugs is not recommended as it increases the risk of adverse effects, but discontinued treatment often causes significant bone loss. For example, discontinuation of denosumab (antibody drug for RANKL) often causes accelerated bone loss in clinics[7,57]. A recent discovery of osteoclast recycling at the end of bone resorption may explain this phenomenon[17,58], but a detailed investigation has been limited due to the difficulty in accessing and monitoring longitudinal osteogenic cellular processes through a mouse calvaria bone. Our in vitro bone model could significantly contribute to elucidating the underlying mechanisms of this clinical situation and facilitate the development of safe and effective pharmacological strategies.

Using our model, researchers can investigate the effects of osteoporosis drugs on osteoclast differentiation, fusion, fission, and apoptosis as well as their impact on osteoblasts and bone remodeling

cycles. This could provide critical insights into how these drugs affect bone remodeling and help identify potential therapeutic targets to prevent bone loss caused by discontinued drug treatments. Conventional drug testing takes 6–14 weeks in preclinical animal studies and 12–24 months in clinical trials, with suboptimal predictive power that can often lead to failures due to unexpected toxicity and lack of efficacy[59]. In vitro humanized bone models that can accurately predict in vivo drug responses could offer a new testing dimension between animal models and clinical studies[60]. We demonstrated the feasibility of this important direction using primary human bone marrow-derived stromal cells that differentiate into osteoblasts and peripheral blood-derived CD14[+] monocytes that differentiate into osteoclasts with M-CSF/RANKL (Fig. 8A–D). Humanized RdBP demonstrated the differential potency between OS680 and Zoledronate, which share the same backbone chemistry with high mineral binding affinity but exhibit different extents of toxicity due to variation in functional groups (Fig. 8E, F). Pamidronate-based OS680 is less toxic than Zoledronate[61], and the conjugation of a fluorescent dye further reduces its toxicity[62]. These results support the efficacy and predictive power of DBP-based platform for evaluating OC-targeting drugs.

In this study, DBP has shown the potential to be used in three distinct bone metabolic assays. First is an osteoblast anabolic assay taking advantage of its ability to induce rapid and substantial mineral deposition. Typically, osteoblast anabolic activities are measured by long-term culture on TCP, as deposition of collagen matrix prior to mineralized nodule formation takes multiple weeks. The pre-existing structural collagen matrix on DBP significantly accelerates osteoblastic mineralized bone formation while more clearly distinguishing anabolic effects. Second is an osteoclast catabolic assay taking advantage of RdBP that closely resembles natural bone ECM while being scalable for manufacturing and retaining optical accessibility. Our studies confirmed that RdBP carries similar functions to support osteoclast differentiation and mineral resorption to BD, the current gold standard for an osteoclast assay. Third is a bone remodeling assay via leveraging coculture osteoblasts and BMMs on DBP under chemical stimulation, which currently does not exist. The coculture model supports prolonged survival, proliferation, and differentiation of BMMs. These reproducible and scalable in vitro osteogenic assays could significantly improve quantitative functional assays of osteoblastic mineralization, osteoclastic bone resorption, and coculture-based bone remodeling (Fig. 9).

The current DBP-based bone model can be improved in multiple aspects to enhance the predictive power of preclinical bone remodeling assays and drug screening. First, the 2D bone model could be advanced to a 3D bone model, including osteocytes that comprise over 95% of total bone cells and play a major role in regulating bone metabolism[63–66]. Second, the current bone model could be combined with a marrow model to accommodate hematopoietic and stromal cells, precursors of osteoclasts and osteoblasts, respectively[67,68]. Third, mechanical and shear forces could be applied to recapitulate the mechanical regulation of bone metabolism[69,70].

In conclusion, we developed a DBP-based osteogenic assay platform in a standard multi-well plate, which offers several advantages over existing assay platforms. Remineralized DBP by osteoblasts restores the structural and material complexity of bone ECM while retaining optical transparency for longitudinal fluorescent imaging. This allows quantitative monitoring of osteoclast fusion and fission, mineral resorption, and bisphosphonate drug response. By offering a more accurate representation of bone ECM and multicellular processes in a scalable and analytical manner, this platform could contribute to osteoclast targeting drug development and elucidating the mechanism of bone metabolic regulation. DBP-integrated multi-well plates could become a standard for studying bone cell biology and osteogenic assays.

## Methods

### Overview of materials and experimental procedures

All chemicals and materials were purchased from Sigma-Aldrich or Fisher Scientific unless otherwise specified. Animal care and experiment procedures were approved by the Institutional Animal Care and Use Committee of the University of Massachusetts Amherst, adhering to federal, state, and local guidelines.

### Preparation of the demineralized bone paper

Bovine femurs were obtained from a slaughterhouse or local grocery stores. A midsection of the femur containing mostly compact bone was demineralized through the rapid demineralization process that we devised previously[29]. Briefly, after removing outer connective tissue and dissolving inner marrow fat in a 1:1 methanol-chloroform solution, the bovine compact bone block was demineralized in 1.2 N hydrochloric acid (HCl) solution with cyclic pressure up to 4-bar with a 10-second on/off interval while replacing HCl solution each day for 5–7 days. The remaining mineral in the bone was checked by quick X-ray scanning (IVIS Spectrum-CT). The demineralization process was repeated until complete mineral removal was confirmed by an X-ray scan.

A fully demineralized bone block was cryosectioned within the range of 10–200 µm; the 20-µm thickness was used in this study (Cryosta NX70). The sectioned demineralized bone matrix was washed overnight with 1% sodium dodecyl sulfate (SDS) solution to remove the remaining cellular materials. Decellularization was confirmed by nucleus DAPI staining. Decellularized DBPs were then washed with deionized water six times and stored in 70% ethanol at 4 °C. The DBP was further tailored into a circular shape using a biopsy punch (D = 6 mm) to fit in 96-well plates. To visualize the DBPs, they were stained with Rhodamine (1 mM) for 30 min and washed with deionized (DI) water.

### Characterization of demineralized bone paper

Mechanical stiffness: DBPs with a thickness of 20 µm were placed in a mechanical testing machine (ElectroForce 5500, TA Instruments). The samples were stretched at a rate of 0.4 mm/s at room temperature, while the tensile force and displacement of the grips were continuously monitored and recorded using XEI software (TA Instruments) until failure occurred. A stress-strain curve was generated in Microsoft Excel, and from this curve, Young's modulus and stiffness of each sample were calculated.

Biochemical integrity of collagen fibers: A fluorescent 5-FAM conjugated collagen hybridized peptide (CHP) (3Helix) was used to assess the biochemical integrity of collagen fibers in DBPs. Heated DBPs were prepared as a positive control with denatured collagen by submerging DBPs in an 80 °C water bath for 1 min. Before the CHP binding test, a CHP stock solution was incubated in a water bath at 80 °C for 5 min to dissociate trimeric strands into monomeric strands. The heated solution was briefly cooled on ice for 30 s. Intact and heated DBPs were incubated overnight in 10 µM of monomeric CHP solution at 4 °C. The CHP bound on damaged collagen fibers was observed under a fluorescence microscope (EVOS). Furthermore, collagen fibers were observed using a resonant scanning multiphoton microscope (Nikon A1MP). Collagen fibers were excited at 810 nm for second harmonic generation (SHG) multiphoton microscopy.

Optical transparency: DBPs with thicknesses of 20, 50, and 100 µm, RdBP with a thickness of 20 µm, and BD with a thickness of 100 µm were placed in a 96-well plate. The absorbance was measured at a wavelength of 600 nm using a microplate reader (Synergy 2, Bio Tek). The relative optical transparency of the DBP, RdBP, and BD was determined by setting the absorbance of the empty well as 100%.

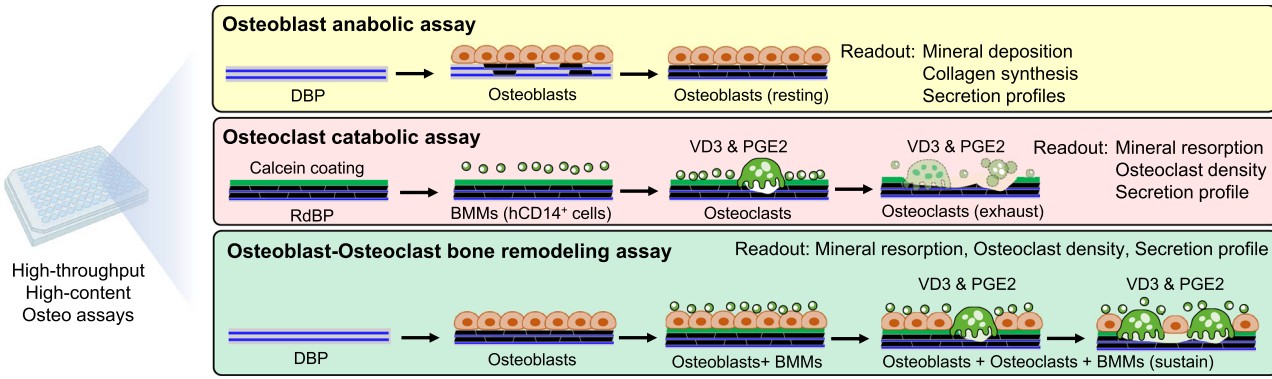

**Fig. 9 | Proposed DBP-based three different osteo assay platforms.** The 96-well plate image has been obtained from BioRender.com.

## Primary murine osteogenic cell isolation, expansion, and differentiation

DsRed mice were obtained from B. Osborne (UMass-Amherst). C57BL/6 mice were obtained from the Jackson Laboratory (000664). Male and female mice aged between 4 to 8 weeks were randomly selected for each batch of the experiment. Primary osteoprogenitors were isolated from the mouse femur and tibia of DsRed mice. After removing marrow via quick centrifugation, the remaining bones were chopped into 1-2 mm in size, then incubated in α-MEM (supplemented with fetal bovine serum (FBS) (10 %) and penicillin and streptomycin (P/S) (1%)) with collagenase (800 U). After overnight incubation, bone chips were washed with phosphate buffered saline (PBS) 3 times and cultured in fresh α-MEM. The culture was not disrupted for 5 days for migrating out osteogenic cells from the collagenase-treated bone chips. Retrieved osteogenic progenitors were expanded on TCP with α-MEM. Passage 3–5 cells were used in experiments. For osteoblast differentiation, $1 \times 10^5$ of osteoprogenitors were seeded on TCP (Corning), RdBP, HP (Corning), BP (Lonza), and BD (Boneslices) and cultured in osteogenic differentiation medium composed of α-MEM with β-glycerophosphate (10 mM), and L-ascorbic acid (200 μM).

## Human bone marrow stromal cell isolation, expansion, and differentiation

Fresh human bone marrow aspirate (50 mL) was purchased from Lonza (age 25–45 male and female donors). Mononuclear cells were isolated by density gradient centrifugation with Ficoll. Isolated mononuclear cells were cultured on TCP using α-MEM supplemented with FBS (10 %), human fibroblast growth factor (5 ng/ml) and P/S (1%). Emerging bone marrow stromal cell (BMSCs) colonies were expanded for 3 rounds at a 1:5 split ratio in T-175 flasks and then cryopreserved. At this stage of BMSCs will be considered passage 0. Less than passage 5 cells were used in experiments. Human BMSCs were cultured in osteoblast differentiation medium composed of α-MEM supplemented with P/S (1 %), FBS (10 %), and β-glycerophosphate (10 mM), L-ascorbic acid (200 μM), and dexamethasone (100 nM).

## Primary murine bone marrow mononuclear cell isolation and differentiation

GFP mice were obtained from the Jackson Laboratory (003291). Male and female mice aged between 4 to 8 weeks were randomly selected for each batch of the experiment. Bone marrow cells were retrieved from the femur via quick centrifugation and plated on a 10 cm petri dish with α-MEM supplemented with FBS (10%) and P/S (1%) to separate adherent stromal cells. After overnight incubation, suspension bone marrow monocytes (BMMs) were collected and used without expansion. For BMM single culture differentiation, $1 \times 10^5$ of BMMs were seeded on TCP, RdBP, HP, BP, MD, and BD with M-CSF (20 ng/mL) and RANKL (40 ng/mL). MD was prepared by subjecting BD to a heat treatment at 500 °C for 5 h, a process designed to thermally decompose the collagen within the BD.

## Human CD14⁺ monocyte cell isolation and differentiation

Peripheral blood mononuclear cells (PBMCs) were isolated from human whole blood by density gradient centrifugation with Ficoll. CD14⁺ monocytes were further isolated from the PBMCs using anti-human CD14 antibody for CD14⁺ cell selection (MagniSort system, 8802-6834, Invitrogen). Antibodies were validated according to the manufacture's description. CD14⁺ cells were placed on a T-25 tissue culture flask with α-MEM supplemented with FBS (10%), P/S (1%) and M-CSF (20 ng/ml) and cultured for 1 week. Before osteoclast differentiation, the colony formation of CD14⁺ cells were confirmed using an optical microscope. $1.5 \times 10^3$ CD14⁺ cells/mm² were introduced on TCP, HP and RdBP with medium composed of α-MEM supplemented with FBS (10%), P/S (1%), human M-CSF (20 ng/mL), and human RANKL (40 ng/mL).

## Mouse Osteoblast-Osteoclast coculture

In advance of the coculture, $5 \times 10^4$ of osteoblasts were cultured on TCP, BD and DBP with osteogenic medium for 1 week. Next, $1 \times 10^5$ of BMMs were introduced into the wells with a growth medium composed of α-MEM supplemented with FBS (10 %), P/S (1 %), vitamin D (VD3) (10 nM), and prostaglandin E2 (PGE2) (1 μM).

## Decellularization of remineralized bone paper

Osteoblasts were cultured on DBP with an osteogenic medium for 1 week. The surface-covered osteoblasts were decellularized using SDS (1%) for 30 min. Afterward, the DBP was thoroughly washed with PBS at least five times to remove any residual SDS and lysed cells. Decellularization was confirmed under a fluorescence microscope by verifying the complete removal of the reporter fluorescence protein.

## Characterization of osteoblasts

Osteoblast number: Osteoblasts were fixed with paraformaldehyde (4%) for 15 min and then stained with DAPI for 30 min to label the cell nuclei. Images of the stained nuclei were acquired using a fluorescent microscope (EVOS). To quantify the number of osteoblasts, the DAPI-stained images were analyzed using EVOS Analysis software, which provides both manual and automated cell counting tools.

Osteoblast alignment: Osteoblasts were fixed with paraformaldehyde (4%) and the cytoskeletal actin filaments were stained with Alexa Fluor 488 or 568 phalloidin, while cell nuclei were stained with DAPI. Actin staining was observed using a fluorescence microscope (EVOS). The angle tool function of ImageJ was used to measure the cell alignment angles. Cell alignment angles on TCP, DBP, HAP, BP and BD were measured with respect to the horizontal line and the average collagen alignment of DBP. The 0° angle was set based on the

collagen alignment angle of DBP. A circular diagram was generated in MATLAB using 40 measurements from 3 different samples.

## Characterization of osteoclasts

Osteoclast cell morphology: Osteoclasts were fixed with paraformaldehyde (4%) and then the cytoskeletal actin filaments were stained with Alexa Fluor 488 phalloidin, while cell nuclei were stained with DAPI. Osteoclast cell morphology was observed using a fluorescence microscope. The size of osteoclasts was manually measured in ImageJ.

TRAP activity: Osteoclasts were fixed in a formaldehyde (4%) and subsequently rinsed three times with DI water. To assess osteoclast differentiation, a TRAP detection kit (387A, Invitrogen) was used following the manufacturer's recommended protocol. The stained cells were then examined using an optical microscope.

## Time-lapse imaging and quantitative imaging analysis

Osteoblasts (DsRed) and osteoclasts (eGFP) were cocultured on DBP with VD3 and PGE2 stimulation. Time-lapse live cell imaging was performed using an inverted fluorescent microscope with a 10× objective lens (LumaScope 720, Etaluma) that operates inside a $CO_2$ incubator. Time-course changes of osteoclast size, number, fusion, fission, and apoptosis were manually tracked and quantified in ImageJ. Apoptotic osteoclasts were individually tracked to measure the life span of osteoclasts. Automatic quantification of osteoclast number was demonstrated using CellProfiler. The accuracy of the results was validated through a direct comparison with manually collected data.

## Determining OPG and RANKL secretions of osteoblasts

Osteoblasts conditioned medium were collected and analyzed OPG and RANKL secretion using an enzyme-linked immunosorbent assay (ELISA) kits for mouse OPG (DY459, R&D systems) and mouse RANKL (DY462, R&D systems). Antibodies were validated according to manufacturer's description. Conditioned media samples were taken at different time points during the culture period and diluted at 1:30 in reagent diluent to ensure that the concentrations of OPG fell within the detection range of the assay. The ELISA assay was then used to measure the concentrations of OPG and RANKL in the conditioned medium.

## RNA sequencing and data analysis

Mouse osteoblasts were cultured on TCP and DBP for 2 weeks with osteogenic medium. Total RNA from mouse osteoblasts on TCP and DBP was isolated using a PureLink RNA mini kit. We followed the manufacturer's general guidelines to get the purified RNA. RNA purity was verified using a microplate reader (Synergy 2, BioTek). Then, RNA sequencing and data analysis were conducted at the Beijing Genomics Institute (BGI) (Shenzhen, China). The sequencing was performed on DNBSEQ (DNBSEQ Technology) platform. SOAPnuke (BGI) software was used for sequencing data filtering. Sequencing reads were mapped by Hierarchical Indexing for Spliced Alignment of Transcripts (HISAT) software. Further sequencing data analysis was processed in Dr.Tom system (BGI). Transcripts per million (TPM) value was used for the expression heatmap. Genes that were upregulated or downregulated were selected based on a criterion of Q-value < 0.05 and 0.5 <| Log2fc |.

## Cell apoptosis assay via Caspase-3/7 detection

To detect the cell apoptosis, Caspase-3/7 green detection reagent (Invitrogen) was treated during osteoblast-osteoclast coculture. The 10x stock reagent was diluted in a growth medium composed of α-MEM supplemented with FBS (10%), P/S (1%), VD3 (10 nM), and PGE2 (1 μM). This solution was directly added to the cells. To distinguish the Caspase-3/7 green signal, non-fluorescent osteoblasts from C57BL/6 mice were cocultured with MNCs from DsRed mice. Fluorescence from apoptotic cells was observed via time-lapse imaging that was performed inside a $CO_2$ incubator.

## Characterization of mineral metabolism

Mineral deposition: Osteoprogenitors were cultured on TCP and DBP for up to 3 weeks with the osteogenic differentiation medium. At the end of the experiment, osteoblasts were fixed with paraformaldehyde (4%) for 5 min and washed 3 times with DI water. The mineral deposition was characterized using fluorescent calcein. For fluorescent calcein staining that has a high binding affinity to calcium minerals, fixed samples were incubated with calcein (10 μM) in calcium-free PBS for 30 min. Minerals stained with calcein were imaged with a fluorescence microscope (EVOS). The fluorescence intensity of calcein bound to minerals was quantified with ImageJ. Mineralized collagen fibers by osteoblasts were imaged using a resonant scanning multiphoton microscope (Nikon A1MP) with a 25× objective lens via second harmonic generation, excited at 810 nm.

Mineral resorption: Mineralized DBP were stained with calcein (10 μM) in calcium-free PBS for 30 min and washed 3 times with PBS before imaging. Time-course bone mineral resorption by osteoclasts was observed under a fluorescent microscope using a 10× objective lens (LumaScope 720, Etaluma) inside a $CO_2$ incubator. The resorbed mineral area was distinguished by decreasing local fluorescence intensity, where mineral resorption made the green fluorescence darker. The resorbed mineral area was quantified using ImageJ, and the resorption rate was calculated by plotting the time-course change of the resorbed area over time. Osteoclastic bone resorption on BD was observed by toluidine blue staining that binds to acidic components of the bone matrix and cell nucleic acids.

## Bisphosphonate treatment

Osteosense (OS680, PerkinElmer) and zoledronate (Zol) stock solution (1 mM) in PBS were diluted in a growth medium with FBS (10%) and P/S (1%). Then, mouse osteoblasts on DBP and BD were incubated with OS680 (5 μM) or Zol (100 nM) for 24 h. For BP testing with human osteoclasts only, RdBP was incubated with OS680 (5 μM) or Zol (5 μM). Bisphosphonate molecules bind to the deposited mineral on DBP and BD. Unbound BP molecules were washed with PBS. To investigate the antiresorptive effect of BPs, further osteoclast culture was performed after bisphosphonate treatment. Confocal microscopy (Zeiss Cell Observer SD) and fluorescence microscope (EVOS) were used for imaging the OS680-bound mineral layer and osteoclasts.

## Statistics and reproducibility

All measurements were obtained at least in triplicate and are presented as means ± standard deviation (SD). *P*-values were determined using unpaired two-tailed Student's *t* tests and are denoted in the figures. A *p*-value of less than 0.05 was deemed to indicate statistical significance. Representative images in the figures were confirmed by reproducing the results in two to three independent experiments. No data points were excluded from the analysis. Animals were allocated to experimental groups on a random basis.

## Reporting summary

Further information on research design is available in the Nature Portfolio Reporting Summary linked to this article.

## Data availability

The RNA sequencing data generated in this study have been deposited in the NCBI Sequence Read Archive under accession code PRJNA104146. Source data are provided with this paper.

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

## Acknowledgements

The authors thank and acknowledge the University of Massachusetts Amherst Light Microscope Facility. We thank Hyejin Yoon for helping with the fabrication of demineralized bone, Jun-Goo Kwak for conducting radiographic imaging, and Ju-Ho Lee for manufacturing cyclic pressure chambers. This work was supported by the National Cancer Institute (R00CA163671, R01CA237171) and the National Science Foundation (1944188). Y.P. was supported by an NSF Research Traineeship (1545399).

## Author contributions

Y.P. and J.L. conceived of and designed the experiments. Y.P. carried out the experiments. Y.P. and J.L. analyzed and interpreted the results, designed the figures, and wrote the manuscript. T.S. analyzed and interpreted the results of RNA sequencing. J.L. supervised the research project and directed the overall research.

## Competing interests

The authors declare no competing interests.
