## [Peer Review File · Nature Communications]

Functional and analytical recapitulation of osteoclast biology on demineralized bone paperREVIEWER COMMENTS

Reviewer #1 (Remarks to the Author):

The authors reported using an osteoid-inspired biomaterial – the demineralized bone paper (DBP) to study osteoclast biology. This technology has been successfully used as a trabecular bone organoid model to study bone remodeling (Sci Adv 2021), and the extension of the use of DBP for osteoclast research is logical and practical. Compared to several standard culture systems, DBP shows promising advantages in recapitulating the in vivo bone niche and produces more physiologically relevant osteoclasts, which have normal size, life span, and optimal responses to hormones and drugs. Largely, the experiment is well-designed and straightforward, and the analysis is appropriate. The high-quality imaging data and movies are convincing. And this biomaterial will likely have a broad usage in the field of bone and osteoclast biology and medicine as articulated in the discussion.

There are some moderate or minor points need to be addressed:

1. The impact of this work can be enhanced significantly if authors could demonstrate that this DBP system can be coupled with other technologies (such as flow cytometry and cell harvested from DBP for molecular biology or sequencing work, etc) for mechanistic studies.
2. What are other ECM components present in the DBP? Do they contribute to the function of DBP? Can osteoclasts or their progenitors attach to DBP without the pre-seeded osteoblast?
3. Does the thickness or orientation of the DBP affect the growth and properties of osteoblasts, osteoclasts, or co-culture?
4. Fig 1. The table below panel B and C has no description in the figure legend, and the meaning of those Xs, Os, and triangles in the table are not explained.
5. “Mononuclear cells” are used in figure 1A, while “BMMs” is used in Fig. 4B and 6C, and “BM-MNCs” is used in Fig 5A. And “BMMs” is used throughout the methods. The terminology should be unified as possible.
6. Descriptions of movies should be included, especially since some of them are not labeled with the corresponding materials (Mov 2, 5, 6).
7. Fig. 5, OPG and RANKL in culture medium were quantified. This result should be normalized with the osteoblast number on each surface because this difference can be due to the proliferation/survival of osteoblasts in VD3 and PGE2 treatments on various surfaces.

Reviewer #2 (Remarks to the Author):

This manuscript describes a novel approach to 3D culture systems for assessing and visualizing bone cell function and dynamics. This new model involved using a small thick disc of bovine tissue which has been demineralized and de-cellularised. It provides opportunity to use 3D in vitro systems to assess the impact of different settings on bone cell behaviour and function and could help advance the field of bone biology in a number of ways. It is well written and data and figures are of good quality, however for

publication in this journal a number of clear limitations should be considered and addressed by the authors as outlined below:

- 1) When comparing DBP to other standard systems, the authors examine osteoclast formation and fate. It is strongly advantageous that the DBP limits the formation of abnormally large/giant osteoclasts as this does not happen in vivo. Can it be confirmed that the DBP or RdBP set-ups do not just delay this formation of giant osteoclasts and if more time was allowed this would occur?
- 2) In addition in these experiments, reference is often made to fission events with a few movies showing these, the authors indeed suggesting that DBP RdBP system supports more normal rates of fission preventing the formation of large osteoclasts – It would be very valuable to have fusion/fission events quantified in each set up to support this? This is also important to the withdrawal of VD3/PEG2 experiments. Did the cells undergo fission or apoptosis or both during this phase? Supplementary movie 6 shows a very different time frame for each set up, TCP is much shorter. Is this because it happens much quicker in TCP? Furthermore, the movies showing fission events could be more clearly annotated to show these.
- 3) For figure 5F and supplementary movie 7, what happened to the osteoclast only cells once resorption had stopped much earlier? Did they get large and die? Or just reduce in their activity?
- 4) In regards to the experiments with Bisphosphonates, how was apoptosis confirmed in the BP treated cells? There are a number of kits/assays one can use to visualize apoptosis – such as caspase kits. It is important that apoptosis is definitively defined compared to fission here. Supplementary movie 8 is difficult to be certain the events circled are fission as the time lapse is too staggered, can these be zoomed in on and slowed down and membrane tracked some how/they look more like apoptosis to me....this really should be made more clear. In addition, tracking of nuclei would also assist with this distinction.
- 5) In regards to OS680 – is this an active BP? There are a number of studies which suggest that fluorescent BP's can have reduced anti-prenylation capacity <https://www.ncbi.nlm.nih.gov/pmc/articles/PMC5117111/> Authors should comment on this with the OS680 probe as it may impact their interpretation of their data. Could a prenylation assay be performed to examine the difference to zol used in the study?
- 6) The BP study was also only performed on the DBP. A comparison to how DBP performs in this context when compared to say BP or HA would be very useful to confirm it is more physiologically relevant for other drug effect analyses.
- 7) There is mention of use of the CellProfiler for data generation but this is not described in detail in the methods.
- 8) In the discussion, there is a comment that this is the first in vitro demonstration of BP action during physiologically relevant bone remodelling. This is not really the case, although not as complex, this has been examined before <https://pubmed.ncbi.nlm.nih.gov/18325866/> This work should be referred to here, and instead suggests that the current work builds on this.
- 9) Finally, the lack of human cell work in this paper limits its impact in the field. It is suggested this system could be used for human cell investigations however this has not been tested yet. For publication at this journal level, some human work validating this potential should be included.

Minor edits:

Abstract; "...treatment demonstrated significantly reduced the number and lifespan..... does not make sense, delete demonstrated?

Intro: Line 43-44 should read: “however recent intravital imaging studies of mouse calvaria and tibia (ref 17)...”

Figure 1 – what do symbols in table mean? O X etc

Revision of NCOMMS-23-17864A-Z for Nature Communications
(Black: comment, Blue: response, Purple: Newly updated results)

Response to Reviewer #1:

Overall comment

The authors reported using an osteoid-inspired biomaterial – the demineralized bone paper (DBP) to study osteoclast biology. This technology has been successfully used as a trabecular bone organoid model to study bone remodeling (Sci Adv 2021), and the extension of the use of DBP for osteoclast research is logical and practical. Compared to several standard culture systems, DBP shows promising advantages in recapitulating the in vivo bone niche and produces more physiologically relevant osteoclasts, which have normal size, life span, and optimal responses to hormones and drugs. Largely, the experiment is well-designed and straightforward, and the analysis is appropriate. The high-quality imaging data and movies are convincing. And this biomaterial will likely have a broad usage in the field of bone and osteoclast biology and medicine as articulated in the discussion.

Response: We appreciate the reviewer 1’s positive assessment of our DBP-based bone modeling.

There are some moderate or minor points need to be addressed:

Comment 1: The impact of this work can be enhanced significantly if authors could demonstrate that this DBP system can be coupled with other technologies (such as flow cytometry and cell harvested from DBP for molecular biology or sequencing work, etc.) for mechanistic studies.

Response: In this revision, we include two comparative RNA-sequencing results: (i) osteoblasts on tissue culture plastic (TCP) and DBP (**Supplementary Figure 1**) and (ii) osteoblasts on DBP with and without vitamin D3 and prostaglandin E2 stimulation (**Fig.5C-E**) (N=3 per group). The RNA-seq results substantiate extensive lists of up and down-regulated genes. The original Excel file of RNA sequencing results is also attached as **Supplementary Data**.

Supplementary Figure 1. Comparative osteoblast gene expression profiles between TCP and DBP. (A) RNA-sequencing heatmap comparing gene expression profiles of osteoblasts between TCP and DBP for selected genes with the Q-value lower than 0.01 (total: 4,654 genes). Distinct and overlapped gene numbers between TCP and DBP. **(B)** The list of top 30 up- and down-regulated genes between TCP and DBP culture. (up-regulated: $\text{Log}_2\text{fc} > 1$, down-regulated: $\text{Log}_2\text{fc} < -1$).

Figure 5. (C) (i) Comparative RNA-seq heat-map of OBs on DBP between resting and VD3/PGE2 stimulated states for selected genes having the Q-value lower than 0.01 (total: 5,249). (ii) Distinct and overlapped gene numbers between resting and activated osteogenic cells. **(D)** A volcano plot of 128 selected bone remodeling-related genes (up-regulated: $\log_2(\text{fold change}) > 0.5$, down-regulated: $\log_2(\text{fold change}) < -0.5$). **(E)** List of 128 genes and their profiles of up- and down-regulation, and non-significant changes (different colors to divide them further into high or low: $1 < |\log_2(\text{fold change})|$, moderate: $0.5 < |\log_2(\text{fold change})| < 1$). (* $p < 0.05$, ** $p < 0.01$, ns, not significant)

Supplementary Data

Supplementary Data: RNAseq analysis Excel file.

Comment 2: What are other ECM components present in the DBP? Do they contribute to the function of DBP? Can osteoclasts or their progenitors attach to DBP without the pre-seeded osteoblast?

Response: DBP primarily consists of collagen that retains the intrinsic biochemical and structural intricacies of the bone matrix, which was confirmed using collagen-hybridized peptide and second harmonic generation imaging (**Fig.1C**). Given prolonged treatment (5-7 days) with hydrochloric acid under cyclic hydrostatic pressure, we expect other biomolecules and growth factors embedded in the bone matrix are negligible.

In this revision, we extended our investigation to better understand why DBP effectively recapitulates in vivo osteoclast morphology and functions by using systematically varied biomaterial platforms. For synthetic materials, we used TCP, collagen-coated TCP, and hydroxyapatite-coated TCP. For the bone ECM, we utilized bone disc, DBP, and mineral discs (thermally-decomposed bone discs) (**Fig.4C**). Osteoclasts on synthetic substrates displayed a spread morphology, while those on bone ECM substrates showed a compact, circular morphology (**Fig.4D**). These findings underscore the importance of preserving the natural structure of bone ECM in regulating the osteoclast fusion, while minerals do not appear to be crucial for facilitating osteoclast differentiation.

Endpoint actin staining highlighted the significance of actin ring structure and position in regulating osteoclast fusion and fission (**Fig.4G**). Without bone ECM, osteoclasts formed a thin actin ring along the cell boundary. When this belt-like actin encountered another actin ring from an adjacent osteoclast, it readily fused, promoting the outward expansion of the cell body. In contrast, osteoclasts on bone ECM substrates developed a dense compact actin ring, typically 20-30 μm inner side of the cell periphery, which might make fusion more challenging. These actin ring configurations align with prior observations of osteoclasts when they engage in pit- or trench-type resorption on bone surfaces. Preserving the integrity of bone ECM, either collagen or mineral, seems essential for faithfully reproducing the appropriate osteoclast phenotype.

Figure 1. (C) The biochemical integrity of collagen in DBP was confirmed using (i) fluorescent dye-conjugated collagen hybridizing peptides (CHP) that specifically bind to denatured collagen fibrils and (ii) second harmonic generation (SHG) imaging that visualizes intact collagen fibrils under multiphoton microscopy. Heated DBP with denatured collagen was used as a positive control.

Figure 4. (C) Preparation of mineral disc (MD) and its surface morphology under scanning electron microscopy imaging.

(D) Representative fluorescent images of mature osteoclasts (green: phalloidin, blue: DAPI) on seven different substrates

Figure 4. (G) (i) Schematic of generalized OC actin ring structure on TCP-based substrates (TCP, Col-TCP, HP) and bone ECM-based substrates (MD, BD, DBP, RdBP). (ii) Quantitative comparison of a ratio of actin ring and osteoclast boundary (n=20).

Comment 3: Does the thickness or orientation of the DBP affect the growth and properties of osteoblasts, osteoclasts, or co-culture?

Response: We have not observed any change in their activities depending on the thickness of the DBP. In this study, the 20 μm thickness of DBP was chosen, a decision grounded in its similarity to the mineralization depth of osteoblast observed in our previous research (Science Advances 2021). Furthermore, this thickness offers favorable mechanical stiffness for easy handling and provides adequate transparency for subsequent imaging analysis.

Regarding the collagen orientation of DBP, osteoblast adhesion aligns with the underlying collagen direction. Currently, we are investigating whether the collagen orientation of DBP affects osteoblast mineral deposition and subsequent osteoclast mineral resorption. We plan to report these studies in a separate manuscript. We hope the reviewer understands our intention.

Comment 4: Fig 1. The table below panel B and C has no description in the figure legend, and the meaning of those Xs, Os, and triangles in the table are not explained.

Response: Thanks for the comment. We have updated the Figure 1 legend with a clear annotation of the meaning of the symbols (o: applicable, Δ: partially applicable, x: not applicable).

Comment 5: “Mononuclear cells” are used in figure 1A, while “BMMs” is used in Fig. 4B and 6C, and “BM-MNCs” is used in Fig 5A. And “BMMs” is used throughout the methods. The terminology should be unified as possible.

Response: Thank you for the comment. We have unified the terminology as bone marrow monocytes (BMMs) in the revised manuscript.

Comment 6: Descriptions of movies should be included, especially since some of them are not labeled with the corresponding materials (Mov 2, 5, 6).

Response: Thank you for the comment. To give a better understanding, we added more detail descriptions of movies in supplementary information.

Comment 7: Fig. 5, OPG and RANKL in culture medium were quantified. This result should be normalized with the osteoblast number on each surface because this difference can be due to the proliferation/survival of osteoblasts in VD3 and PGE2 treatments on various surfaces.

Response: Thank you for this comment. We normalized OPG and RANKL concentration by the osteoblast number per area (**Fig.5B**). We did not observe the significant change in osteoblast viability and proliferation before and after VD3/PGE2 treatment. The trend of secretion profiles was the same as the previous data without normalization. In this revision, we include the characterization of BD.

Figure 5. (B) Comparative RANKL and OPG secretion profiles of OBs on TCP, DBP, and BD (i) before and (ii) after VD3/PGE2 stimulation (n=4).

Response to Reviewer #2:

Overall comment

This manuscript describes a novel approach to 3D culture systems for assessing and visualizing bone cell function and dynamics. This new model involved using a small think disc of bovine tissue which has been demineralized and de-cellularised. It provides opportunity to use 3D in vitro systems to assess the impact of different settings on bone cell behaviour and function and could help advance the field of bone biology in a number of ways. It is well written, and data and figures are of good quality, however for publication in this journal a number of clear limitations should be considered and addressed by the authors as outlined below:

Response: Thank you for the positive assessment of our work. We appreciate your time and constructive feedback that we addressed. Hope you look favorably at these updates.

Comment 1: When comparing DBP to other standard systems, the authors examine osteoclast formation and fate. It is strongly advantageous that the DBP limits the formation of abnormally large/giant osteoclasts as this does not happen in vivo. Can it be confirmed that the DBP or RdBP set-ups do not just delay this formation of giant osteoclasts and if more time was allowed this would occur?

Response: We conducted the osteoclast culture on DBP for two weeks, and there was no increase in osteoclast size (**Fig.4E**). As mentioned above, we conducted systematically varied biomaterials for osteoclastogenesis. These results confirmed that bone ECM structure (collagen or mineral) is critical in regulating osteoclast fusion. One identified mechanism is distinct actin ring structure and position.

Figure 4. (E) Quantified osteoclast size after 7 days of RANKL/M-CSF stimulated culture on different substrates. The inner panel shows osteoclast size after 14 days of stimulated culture on RdBP (n=9-15)

Comment 2-1: In addition, in these experiments, reference is often made to fission events with a few movies showing these, the authors indeed suggesting that DBP RdBP system supports more normal rates of fission preventing the formation of large osteoclasts – It would be very valuable to have fusion/fission events quantified in each set up to support this?

Response: Thank you for your feedback. We quantitatively assessed osteoclast fusion, fission, and apoptosis by analyzing time-course fluorescent imaging of BMMs on DBP (**Fig.6D**) and TCP (**Fig. 6E**). Osteoclast fusion on DBP was reduced about 4 times compared to osteoclasts on TCP. In addition, osteoclast fusion on TCP continued more than 24 hours after VD3/PGE2 withdrawal, whereas osteoclast fusion on DBP stopped about 6 hours after VD3/PGE2 withdrawal. These differences are likely attributed to distinct actin ring configuration and position between TCP and DBP.

Figure 6. (C) Representative images of osteoclast fusion and fission (blue: DAPI). (D) (i) Time-course measurement of osteoclast size change during a VD3/PGE2 stimulation-induced bone remodeling cycle of osteoblasts and BMMs coculture on DBP (n=10). (ii) Time-course quantified cellular events of osteoclast fusion, fission, and apoptosis and (iii) comparative cellular event ratios during active remodeling periods on DBP. (blue: fusion, red: fission, black: apoptosis) (E) (i) Time-course quantified cellular events of osteoclast fusion, fission, and apoptosis and (ii) comparative cellular event ratios during active remodeling periods on TCP.

Comment 2-2: This is also important to the withdrawal of VD3/PEG2 experiments. Did the cells undergo fission or apoptosis or both during this phase?

Response: Osteoclasts undergo fission and apoptosis on TCP and DBP, even during VD3/PGE2 stimulation. When VD3/PGE2 withdrawal, osteoclast fusion occurred significantly more on DBP than TCP, whereas relative apoptosis event was comparable between TCP and DBP.

Comment 2-3: Supplementary movie 6 shows a very different time frame for each set up, TCP is much shorter. Is this because it happens much quicker in TCP? Furthermore, the movies showing fission events could be more clearly annotated to show these.

Response: The answer is yes. Osteoclast apoptosis happens quicker in TCP once withdrawn VD3/PGE2. Osteoclast on TCP also usually showed frequent cell apoptosis due to their abnormal fusion and giant cell size. We updated the quality of the supplementary movies showing the zoomed-in osteoclast fusion and fission (**Supplementary movies 6 & 7**) with descriptions.

Supplementary Movie 6. Osteoclast fusion during OB-OC coculture on DBP.
eGFP osteoclast precursors fused during OB-OC coculture with VD3/PGE2 stimulation.

Supplementary Movie 7. Osteoclast fission during OB-OC coculture on DBP.
Cell nuclei were stained with Live Cell DAPI reagent. Multinucleated osteoclasts divided into multiple cells during fission after VD3/PGE2 withdrawal.

Comment 3: For figure 5F and supplementary movie 7, what happened to the osteoclast only cells once resorption had stopped much earlier? Did they get large and die? Or just reduce in their activity?

Response: In our study, osteoclast single culture with RANKL and M-CSF supplement underwent apoptosis following bone resorption, which eventually exhausted BMMs and stopped the bone resorption. This was a distinct difference from the osteoblast-osteoclast co-culture under VD3/PGE2 stimulation, where osteoclasts underwent fission and apoptosis, and BMMs maintained prolonged periods. Our RNA sequencing results explain that the increased secretion of growth factors by VD3/PGE2 stimulated osteoblasts supports the continued presence of BMMs and osteoclastogenesis.

Comment 4: In regard to the experiments with Bisphosphonates, how was apoptosis confirmed in the BP treated cells? There are a number of kits/assays one can use to visualize apoptosis – such as caspase kits. It is important that apoptosis is definitively defined compared to fission here. Supplementary movie 8 is difficult to be certain the events circled are fission as the time lase is too staggered, can these be zoomed in on and slowed down and membrane tracked somehow/they look more like apoptosis to me.....this really should be made more clear. In addition, tracking of nuclei would also assist with this distinction.

Response: Thanks for the practical suggestions! We used a caspase-3/7 dye for fluorescent monitoring of live-cell apoptosis. We provided an additional movie clip (**Supplementary Movie 10**). Here, we used osteoblasts derived from non-fluorescent B6 mice to capture green signals from caspase-3/7 dye. As mentioned above, we updated osteoclast fusion and fission movies. We included live nucleus dye (DAPI) for osteoclast fission to verify the nuclei separation during osteoclast fission (**Supplementary Movie 7**).

Supplementary Movie 10. Caspase3-7 assay during OB-OC coculture on DBP.

DsRed osteoclasts were cocultured with non-fluorescent osteoblasts on DBP with (right) and without (left) OS680 treatment. Apoptotic osteoclasts showed a bright green signal of Caspase3/7 detection reagent.

Comment 5: In regard to OS680 – is this an active BP? There are a number of studies which suggest that fluorescent BP's can have reduced anti-prenylation capacity

<https://www.ncbi.nlm.nih.gov/pmc/articles/PMC5117111/>

Authors should comment on this with the OS680 probe as it may impact their interpretation of their data. Could a prenylation assay be performed to examine the difference to Zol used in the study?

Response: Thank you for this insightful comment! We have not recognized this potential impact as we purchased OS680 from the commercial vendor. Based on our understanding, this probe was first reported in 2001 by Nature Biotechnology, “In vivo near-infrared fluorescence imaging of osteoblastic activity” (Pamidronate-based chemistry). We thought extending a prenylation assay for OS680 (that we purchased but not synthesized) was beyond the scope of this manuscript. Instead, we mentioned the original paper for OS680 and its anti-resorptive potency literature. We hope the reviewer understands our limited response to this comment.

Comment 6: The BP study was also only performed on the DBP. A comparison to how DBP performs in this context when compared to say BP or HA would be very useful to confirm it is more physiologically relevant for other drug effect analyses.

Response: Thank you for the comment! TCP-based substrates, including Col-TCP and HP, did not reproduce the normal size of osteoclasts in vivo. A thin synthetic hydroxyapatite layer on HP was quickly resorbed by osteoclasts, turning the substrate back into TCP. A randomly deposited collagen fiber on Col-TCP did not reduce osteoclast fusion. In BP, the reduction of bone particles attached to TCP by osteoclasts facilitated the quantification of bone resorption. However, BP was impractical for in vitro osteo-assays, as the co-existence of bone particles and TPC complicated the execution of standardized, consistent experiments. For this reason, we moved BP-related results to **Supplementary Figure 4** in this revision. Instead, we focused on how BD is treated with OS680 and the subsequent removal of OS680 from BD by mature osteoclasts. The OS680 adhesion on BD was notably more uniform than on remineralized DBP. The diminished fluorescence observed due to osteoclast action allowed for a comparison between BD and remineralized DBP (**Fig.7C**). However, using BD for in situ time-course imaging posed challenges.

Supplementary Figure 4. Characterization of osteoclastogenesis on bone particle-coated plate (BP). (i) Schematic and (ii) actin-stained images of osteoclasts on BP, and (iii) comparison of osteoclast size on BP, TCP, and RdBP. (iv) Time-lapse microscopic monitoring of osteoclastic mineral resorption on BP. (v) Time-course measurements of osteoclastic mineral resorption area on BP with a bone resorption rate (n=3). (* $p < 0.05$)

Figure 7. (C) 3D confocal microscope images show decreased OS680 fluorescent intensity after OC resorption on (i) BD and (ii) RdBP. (iii) Z-stack image confirms locally decreased OS680 fluorescent intensity where OCs reside.

Comment 7: There is mention of use of the CellProfiler for data generation, but this is not described in detail in the methods.

Response: Thank you for the comment. We have included a detailed process of cell profiler analysis in the supplementary information and method section (**Supplementary Figure 5**).

Supplementary Figure 5. Quantitative imaging analysis algorithm of osteoclast number. eGFP+ large BMMs were identified, and cell nuclei were stained with DAPI. Individual channel images were processed to optimize the intensity and contrast by using CellProfiler. Then, fluorescent signals were identified as objects. An eGFP+ cell including more than 3 nuclei was counted as an osteoclast.

Comment 8: In the discussion, there is a comment that this is the first in vitro demonstration of BP action during physiologically relevant bone remodeling. This is not really the case, although not as complex, this has been examined before <https://pubmed.ncbi.nlm.nih.gov/18325866/>. This work should be referred to here, and instead suggests that the current work builds on this.

Response: Thank you for pointing out this great article. We simply have not recognized this important paper. We have included this reference and updated our statement in the discussion section.

Comment 9: Finally, the lack of human cell work in this paper limits its impact in the field. It is suggested this system could be used for human cell investigations however this has not been tested yet. For publication at this journal level, some human work validating this potential should be included.

Response: Thanks for your comment! We agree with the reviewer's point. In this revision, we demonstrated the feasibility of a humanized DBP-based bone model. For osteoblasts, we utilized human bone marrow-derived stromal cells that differentiate into osteoblasts. For osteoclasts, we used human peripheral blood-derived CD14+ monocytes that differentiate into osteoclasts in the presence of human M-CSF and RANKL. We prepared a humanized RdBP by culturing human osteoblasts on bovine DBP, followed by decellularization. We then demonstrated the differentiation of human osteoclasts on the RdBP. Finally, we evaluated the predictive power of humanized RdBP for bone-targeting drug testing by testing human osteoclastogenesis on OS680 and Zoledronate-targeted humanized RdBP. As expected, the results showed a significantly higher anti-osteoporosis potency of zoledronate (Fig.8).

Figure 8. Human osteoclast responses to bisphosphonate drugs on humanized RdBP. (A) Schematic of human bone marrow stromal cell isolation, expansion, and differentiation into osteoblasts on DBP. After 2 weeks of remineralization in osteogenic differentiation medium, humanized RdBP was prepared by decellularization. **(B)** (i) Human osteoblasts adhere following the collagen fiber orientation on DBP but randomly on TCP. (ii) Alizarin red mineral staining after osteoblast culture on DBP and TCP. (iii) Time-course quantitative measurement of mineral deposition by fluorescent calcein staining (n=6). **(C)** (i) Schematic of human hCD14+ cell isolation and expansion from peripheral blood using magnetic-activated cell sorting system. (ii) Brightfield images of ex vivo expanded hCD14+ cells for 1 week culture with hM-CSF. (iii) hCD14+ cells cultured on TCP, HP, and RdBP for 9 days with hM-CSF and hRANKL to induce osteoclastogenesis. **(D)** (i) Representative images of human osteoclasts on TCP, HP, and RdBP. (green: phalloidin, blue: DAPI) (ii) Characterization of human osteoclast sizes between TCP and RdBP at the end of the stimulated culture (n=20). **(E)** (i) Representative time-course fluorescent images of calcein-coated mineral resorption by human osteoclasts on control, OS680-, and zoledronate-treated RdBP. (ii) Time-course quantitative monitoring of resorption areas. **(F)** (i) Representative fluorescent images of human osteoclasts on control, OS680-, and zoledronate-treated RdBP. (green: phalloidin, blue: DAPI) (ii) Normalized human osteoclast density at the end of 2 weeks stimulated culture (n=5). (* p<0.05, ** p<0.01, ns, not significant)

Comment 10: Abstract; "...treatment demonstrated significantly reduced the number and lifespan..... does not make sense, delete demonstrated?"

Response: Thanks for your comment. We edited the abstract in the revision.

Comment 11: Intro: Line 43-44 should read: "however recent intravital imaging studies of mouse calvaria and tibia (ref 17)...."

Response: Thanks for your comment. We edited the lines as below.

"However, recent intravital imaging studies of mouse calvaria and tibia have revealed that osteoclastogenesis is more complex than previously observed in vitro, with at least two distinct aspects¹⁴."

Comment 12: Figure 1 – what do symbols in table mean? O X etc

Response: Thanks for your comment. We have updated the Fig.1 legend with a clear annotation of the meaning of the symbols (o: applicable, Δ: partially applicable, x: not applicable).

REVIEWERS' COMMENTS

Reviewer #1 (Remarks to the Author):

The only suggestion is to include the sexes and ages of mice used as the source of BMM for RNAseq and other experiments.

Otherwise, the revision is very well done and all questions were satisfactorily addressed.

Reviewer #2 (Remarks to the Author):

The authors have made considerable effort to address the reviewers concerns. They have addressed most of the issues raised however I would like to point out 2 areas which could still be addressed:

1) the movies could be improved with additional annotation, in particular the fusion and fission movies in video 6 and 7. The authors could annotate when the fusion/fission events occur with arrows and highlight cells more clearly. Further, the intensity of the fluorescent signal in these movies is faint, could it be improved so any nanotubes between cells can be seen more clearly?

2) The discussion should comment on how the OS680 was less potent than the Zol used and that this may pertain to it having lower activity or a higher dose needs to be used, this is especially important in light of this publication highlighting the different levels of activities of fluorescent BPs:

<https://pubmed.ncbi.nlm.nih.gov/19032080/>

minor edits:

1) remove this sentence as you have added human data now:

Finally, the humanized DBP-based bone model could be advanced to human osteoblast and osteoclast co-culture, which can greatly increase the translational opportunity of preclinical studies 69, 70.

2) Edit the sentence below:

"...On synthetic substrates, osteoclasts exhibit a belt-like 522 thick actin structure positioned close to the cell periphery, which could facilitate fusion upon - think should be thin.

Revision of NCOMMS-23-17864A-Z for Nature Communications

Response to Reviewer #1:

The only suggestion is to include the sexes and ages of mice used as the source of BMM for RNAseq and other experiments. Otherwise, the revision is very well done, and all questions were satisfactorily addressed.

Response: We have updated the sexes and ages of mice. Thank you for your favorable review.

“Male and female mice aged between 4 to 8 weeks were randomly selected for each bath of the experiment.”

Response to Reviewer #2:

The authors have made considerable effort to address the reviewers concerns. They have addressed most of the issues raised however I would like to point out 2 areas which could still be addressed:

1) the movies could be improved with additional annotation, in particular the fusion and fission movies in video 6 and 7. The authors could annotate when the fusion/fission events occur with arrows and highlight cells more clearly. Further, the intensity of the fluorescent signal in these movies is faint, could it be improved so any nanotubes between cells can be seen more clearly?

Response: We adjusted intensity of the fluorescent signal and included arrows in the supplementary movie 6 and 7, which make them clear and easy to observe fusion and fission processes. Thank you!

2) The discussion should comment on how the OS680 was less potent than the Zol used and that this may pertain to it having lower activity or a higher dose needs to be used, this is especially important in light of this publication highlighting the different levels of activities of fluorescent BPs:

<https://pubmed.ncbi.nlm.nih.gov/19032080/>

Response: We include the comment on how OS680 is less potent than the Zol. Thank you for this careful comment and the pointed reference.

“Humanized RdBP demonstrated the differential potency between OS680 and Zoledronate, which share the same backbone chemistry with high mineral binding affinity but exhibit different extents of toxicity due to variation in functional groups (Fig. 8E-F). Pamidronate-based OS680 is less toxic than Zoledronate⁶², and the conjugation of a fluorescent dye further reduces its toxicity⁶³. These results support the efficacy and predictive power of DBP-based platform for evaluating OC-targeting drugs”

Minor edits:

1) remove this sentence as you have added human data now:

“Finally, the humanized DBP-based bone model could be advanced to human osteoblast and osteoclast co-culture, which can greatly increase the translational opportunity of preclinical studies 69, 70.”

Response: We removed this sentence.

2) Edit the sentence below:

"...On synthetic substrates, osteoclasts exhibit a belt-like 522 thick actin structure positioned close to the cell periphery, which could facilitate fusion upon - think should be thin.

Response: We updated this typo. Thank you for carefully checking the manuscript!